# Synergy between Wsp1 and Dip1 may initiate assembly of endocytic actin networks

Connor J Balzer[1], Michael L James[2], Heidy Y Narvaez-Ortiz[1], Luke A Helgeson[1†], Vladimir Sirotkin[2], Brad J Nolen[1]*

[1]Department of Chemistry and Biochemistry, Institute of Molecular Biology, University of Oregon, Eugene, United States; [2]Department of Cell and Developmental Biology, SUNY Upstate Medical University, Syracuse, United States

**Abstract** The actin filament nucleator Arp2/3 complex is activated at cortical sites in *Schizosaccharomyces pombe* to assemble branched actin networks that drive endocytosis. Arp2/3 complex activators Wsp1 and Dip1 are required for proper actin assembly at endocytic sites, but how they coordinately control Arp2/3-mediated actin assembly is unknown. Alone, Dip1 activates Arp2/3 complex without preexisting actin filaments to nucleate 'seed' filaments that activate Wsp1-bound Arp2/3 complex, thereby initiating branched actin network assembly. In contrast, because Wsp1 requires preexisting filaments to activate, it has been assumed to function exclusively in propagating actin networks by stimulating branching from preexisting filaments. Here we show that Wsp1 is important not only for propagation but also for initiation of endocytic actin networks. Using single molecule total internal reflection fluorescence microscopy we show that Wsp1 synergizes with Dip1 to co-activate Arp2/3 complex. Synergistic co-activation does not require preexisting actin filaments, explaining how Wsp1 contributes to actin network initiation in cells.

*For correspondence:
bnolen@uoregon.edu

Present address: †Department of Biochemistry, University of Washington, Seattle, United States

Competing interests: The authors declare that no competing interests exist.

## Introduction

Arp2/3 complex is an important cytoskeletal regulator that nucleates actin filament networks important in a broad range of cellular processes, including cell motility, differentiation, endocytosis, meiotic spindle positioning, and DNA repair (*Goley and Welch, 2006*; *Hurst et al., 2019*; *Rotty et al., 2013*; *Yi et al., 2011*). Multiple classes of nucleation promoting factors (NPFs), including WASP family proteins (Type I NPFs), cortactin and related proteins (Type II NPFs), and WISH/DIP/SPIN90 (WDS) family proteins, activate the nucleation activity of Arp2/3 complex in response to cellular signals (*Goley and Welch, 2006*; *Wagner et al., 2013*). In vitro, activated NPFs can function independently to stimulate actin filament nucleation by Arp2/3 complex, but in cells, actin networks assembled by Arp2/3 complex frequently contain multiple classes of NPFs with nonredundant roles in actin assembly (*Galletta et al., 2008*; *Murphy and Courtneidge, 2011*; *Sirotkin et al., 2005*). Understanding how distinct NPFs coordinately control Arp2/3 complex to assemble cellular actin networks is critical to understanding actin regulation.

At sites of endocytosis in *Schizosaccharomyces pombe*, Arp2/3 complex nucleates the assembly of branched actin networks that drive invagination of the plasma membrane (*Sun et al., 2019*). The activity of Arp2/3 complex at endocytic sites can be controlled by at least three distinct NPFs: Wsp1, Dip1, and Myo1 (*Sirotkin et al., 2005*; *Wagner et al., 2013*). Analysis of mutant or knockout strains of Wsp1 and Dip1 suggests that activation of Arp2/3 complex by both of these NPFs is required for normal endocytic actin assembly (*Basu and Chang, 2011*; *Sirotkin et al., 2005*; *Wagner et al., 2013*). It is currently unknown how Wsp1 and Dip1 cooperate to assemble functional endocytic actin networks in *S. pombe*, but key biochemical differences between these NPFs have

led to a model for their coordinate activity. Wsp1, the *S. pombe* member of the WASP family NPFs, has a characteristic VCA motif at its C-terminus that is sufficient for activation of Arp2/3 complex (*Higgs and Pollard, 2001*; *Sirotkin et al., 2005*). The CA segment within this motif binds Arp2/3 complex at two sites (*Boczkowska et al., 2014*; *Luan et al., 2018b*; *Padrick et al., 2011*; *Ti et al., 2011*; *Zimmet et al., 2020*), while the V segment binds actin monomers, which Wsp1 must recruit to the complex to trigger nucleation (*Chereau et al., 2005*; *Marchand et al., 2001*; *Rohatgi et al., 1999*). Importantly, the Wsp1-bound Arp2/3 complex must also bind to a preexisting actin filament to stimulate nucleation (*Achard et al., 2010*; *Machesky et al., 1999*; *Smith et al., 2013a*; *Wagner et al., 2013*). This requirement ensures that Wsp1 creates branched actin filaments when it activates Arp2/3 complex, but also means a preformed filament must be provided to seed assembly of the network. Dip1, like the other members of the WISH/DIP/SPIN90 (WDS) family proteins, uses an armadillo repeat domain to bind and activate Arp2/3 complex (*Luan et al., 2018a*; *Shaaban et al., 2020*). Unlike Wsp1, Dip1 does not require a preexisting actin filament to trigger nucleation (*Wagner et al., 2013*). Therefore, Dip1-mediated activation of Arp2/3 complex creates linear filaments instead of branches (*Wagner et al., 2013*). Importantly, the linear filaments nucleated by Dip1-activated Arp2/3 complex can activate Wsp1-bound Arp2/3 complex, which creates new branched actin filaments that activate subsequent rounds of Wsp1-Arp2/3-mediated branching nucleation (*Balzer et al., 2018*). Therefore, by activating Arp2/3 complex without a preformed actin filament, Dip1 kickstarts the assembly of branched actin networks. These biochemical observations have led to a model of how Dip1 and Wsp1 coordinate actin assembly at endocytic sites in yeast. In this model, the role of Dip1 as an NPF is solely to generate seed filaments that initiate the assembly of the endocytic actin network, whereas Wsp1 exclusively functions as a propagator of branched networks once they have been initiated.

Recent live cell imaging data support distinct seeding and propagating roles for Dip1 and Wsp1, respectively. For instance, in *dip1Δ* strains, the rate of initiation of new patches is markedly decreased, but once an endocytic actin network is initiated, it assembles rapidly, suggesting Dip1 is important for seeding but not propagation of the network (*Basu and Chang, 2011*). Further, deletion of the Wsp1 CA segment motif causes failure of endocytic actin patches to internalize, a process thought to be dependent on the propagation of branches (*Sun et al., 2019*). However, some data suggests that the seeding and propagating roles of Wsp1 and Dip1 might overlap. Specifically, biochemical and structural data suggested that the two NPFs might simultaneously bind Arp2/3 complex, so they may potentially synergize to activate nucleation (*Luan et al., 2018a*; *Luan et al., 2018b*; *Wagner et al., 2013*).

Here we show that contrary to the previous model, Wsp1 cooperates with Dip1 to generate seed filaments. We provide evidence that this cooperation is important for initiation of endocytic actin networks in cells. By imaging endocytic actin patch dynamics in *S. pombe*, we find that while Wsp1 is a key biochemical propagator of branched actin networks, it also significantly influences the rate at which new endocytic actin patches are created in *S. pombe*, indicating it plays a role in initiation. Through single molecule total internal reflection fluorescence (TIRF) microscopy along with kinetic assays and modeling, we find that the role of Wsp1 in initiation is likely due to its ability to synergize with Dip1 to activate Arp2/3 complex. Specifically, we show that Dip1 and Wsp1 co-activate actin filament nucleation by Arp2/3 complex in vitro. Unexpectedly, in co-activating the complex with Wsp1, Dip1 converts Wsp1 from a branched to linear filament generating NPF. Co-activation by Wsp1 and Dip1 requires actin monomer recruitment by Wsp1 but does not require a preformed actin filament. As a result, the two NPFs together can more potently create seed filaments for branched network initiation than Dip1 alone. This explains the decreased rate of patch initiation resulting from Wsp1 mutations that block its activation of Arp2/3 complex in cells.

## Results

### Deletion of the WASP CA segment causes a decrease in the patch initiation rate

To test their relative importance in the initiation versus propagation of endocytic actin networks, we measured the influence of Dip1 and Wsp1 mutations on actin dynamics in fission yeast using the endocytic actin patch marker Fim1 labeled with green fluorescent protein (GFP) (*Berro and Pollard,*

*2014*). In wild-type cells, Fim1-marked actin patches accumulate in cortical puncta over ~5 s before moving inward and simultaneously disassembling (*Figure 1A–C*, *Figure 1—video 1*; *Berro and Pollard, 2014*; *Sirotkin et al., 2010*). To quantify actin patch initiation defects, we measured the rate at which new Fim1-marked puncta appeared in the cell (*Figure 1D*). As expected based on previous results, the Dip1 deletion strain showed a significant reduction in the patch initiation rate compared to the wild-type strain (0.030 versus 0.0076 patches/s/$\mu M^2$) and a corresponding decrease in the number of actin patches in the cell (*Figure 1D and E*, *Figure 1—video 2*; *Basu and Chang, 2011*). However, once actin assembly was initiated, Fim1-GFP accumulated more rapidly than in the wild-type strain (*Figure 1B and C*). These observations are consistent with previously reported measurements (*Basu and Chang, 2011*) and suggest that Dip1 contributes to the initiation but not the propagation of branched endocytic actin networks. We note that while *dip1* deletion caused increased Fim1-GFP to accumulate in individual patches (*Figure 1B*), the ratio of the total actin in patches versus total actin in the cell decreased in the *dip1Δ* strain because fewer patches were initiated. Therefore, the reduced rate of actin patch assembly cannot be explained by excess filamentous actin in the patches depleting the concentration of soluble actin monomers in the cytoplasm (*Figure 1—figure supplement 1*).

To investigate the contribution of Wsp1 toward initiation and propagation of the actin networks, we deleted the sequence encoding the CA segment of Wsp1 in the endogenous *wsp1* locus and measured the influence of this mutation on actin dynamics. Deletion of the CA segment prevents Wsp1 from binding or activating Arp2/3 complex (*Marchand et al., 2001*), but leaves intact its WASP-homology 1 (WH1) domain, proline-rich segment, and actin-binding Verprolin-homology motif (V). In the *wsp1ΔCA* mutant the average time between the first appearance of Fim1-GFP and when it reaches peak concentration, which we refer to here as the Fim1 assembly time, increased from 5.1 to 6.7 s, consistent with another recent study (*Figure 1B and C*; *Sun et al., 2019*). In addition, the *wsp1ΔCA* mutation decreased the percentage of actin patches that internalized (*Figure 1F*, *Figure 1—video 3*). These observations are consistent with a role for Wsp1 in the propagation of branched actin during endocytosis. However, to our surprise, we found that *wsp1ΔCA* cells also showed a 40% decrease in the rate of initiation of new actin patches compared to wild-type cells (*Figure 1D*). While this defect is less than observed in *dip1Δ* cells, it suggests – contrary to our initial prediction – that Wsp1 may play a role in initiating new endocytic actin patches. Combining *wsp1ΔCA* and *dip1Δ* in the same strain (*dip1Δ wsp1ΔCA*) did not decrease the actin patch initiation rate more than *dip1Δ* alone (*Figure 1D*, *Figure 1—video 4*). This suggests that Wsp1 may contribute to the Dip1-mediated actin patch initiation pathway rather than acting in a separate parallel pathway for initiation of actin assembly.

## Dip1 and Wsp1 synergize during Arp2/3-mediated actin filament assembly

Previous biochemical and structural data suggested that Dip1 and Wsp1 might simultaneously bind Arp2/3 complex, so they could potentially cooperate to activate nucleation (*Luan et al., 2018a*; *Luan et al., 2018b*; *Wagner et al., 2013*). We reasoned that by directly synergizing with Dip1 to activate Arp2/3 complex, Wsp1 could contribute to actin network initiation. However, how the two NPFs together influence the activity of Arp2/3 complex is uncertain. Previous data showed Dip1 is a more potent activator of Arp2/3 complex than Wsp1, and that mixing both Wsp1 and Dip1 in a bulk actin polymerization assay increased the actin polymerization rate, but the reason for the increase was unknown (*Wagner et al., 2013*). Specifically, because those experiments were carried out at sub-saturating conditions, it was unclear whether Wsp1 and Dip1 activate in independent but additive pathways or alternatively, if the two NPFs synergize in activation of Arp2/3 complex. To test this, we titrated Dip1 into actin polymerization reactions containing Arp2/3 complex and the minimal Arp2/3-activating region of Wsp1, Wsp1-VCA. Dip1 approaches saturation above ~30 $\mu$M, and at this concentration the addition of Wsp1-VCA increased the maximum polymerization rate in the pyrene actin assembly assays ~1.6-fold over reactions with Dip1 alone (*Figure 2A and B*, *Supplementary file 1*). These results suggest that the increased actin polymerization rates in the presence of both NPFs cannot be explained by an additive effect in activating Arp2/3 complex, but instead the NPFs are synergistic.

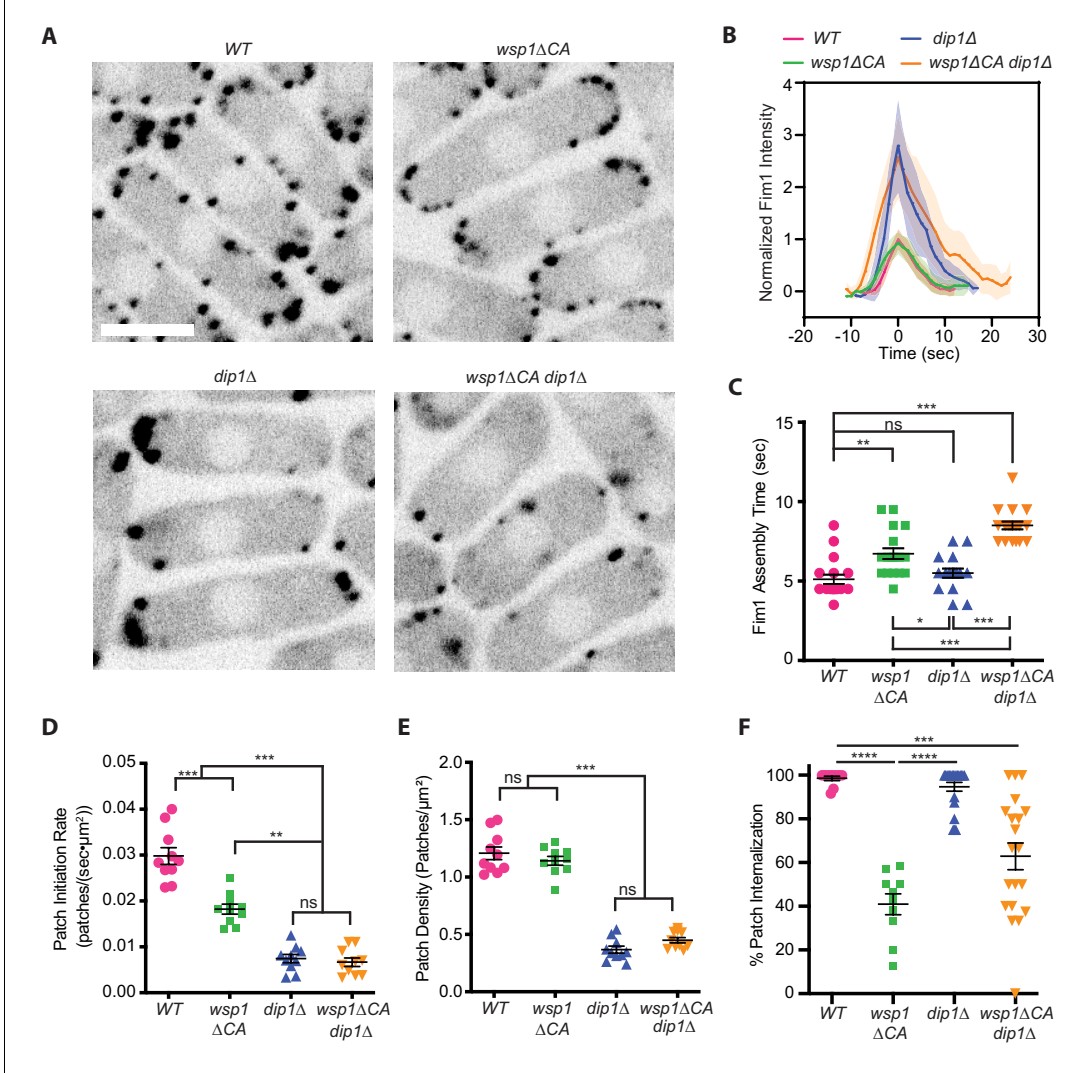

**Figure 1.** Wsp1 plays a role in the initiation of endocytic actin patches. (**A**) Equatorial plane images of Fim1-GFP in *Schizosaccharomyces pombe* cells taken using spinning disk confocal microscopy. Scale Bar: 5 μm. (**B**) Plot showing the relative Fim1-GFP intensity in *S. pombe* mutant endocytic patches over their lifetimes. Traces represent the average of 16–18 endocytic patches. Intensity plots were aligned based on their peak values and normalized based on the peak value of the Fim1-GFP signal in the wild-type strain. Standard deviation is shown as shaded region around each trace. (**C**) Plot comparing the assembly time of Fim1-GFP in endocytic patches in wild-type cells to *wsp1ΔCA*, *dip1Δ*, and *wsp1ΔCA dip1Δ* mutants. Error bars: standard error from 16 to 18 patches. (**D**) Plot comparing the endocytic patch initiation rate in wild-type cells to *wsp1ΔCA*, *dip1Δ*, and *wsp1ΔCA dip1Δ* mutants. Error bars: standard error from 10 cells. (**E**) Plot comparing the endocytic actin patch density in wild-type cells to *wsp1ΔCA*, *dip1Δ*, and *wsp1ΔCA dip1Δ* mutants as determined based on the number of Fim1-GFP-marked cortical puncta. Error bars: standard error from 10 cells. (**F**) Plot showing the percentage of endocytic patches internalized in wild-type and mutant *S. pombe* cells. Error bars: standard error from 10 to 22 cells. p-values: *<0.05, **<0.01, ***<0.001.

The online version of this article includes the following video, source data, and figure supplement(s) for figure 1:

**Source data 1.** Analysis of spinning disk confocal videos of *S. pombe* source data.

**Figure supplement 1.** Nucleation promoting factor (NPF) mutations do not deplete the pool of soluble actin monomers in the cytoplasm.

**Figure 1—video 1.** Spinning disk confocal microscopy video of Fim1-GFP in wild-type *Schizosaccharomyces pombe* cells.
https://elifesciences.org/articles/60419#fig1video1

**Figure 1—video 2.** Spinning disk confocal microscopy video of Fim1-GFP in *dip1Δ Schizosaccharomyces pombe* cells.
https://elifesciences.org/articles/60419#fig1video2

**Figure 1—video 3.** Spinning disk confocal microscopy video of Fim1-GFP in *wsp1ΔCA Schizosaccharomyces pombe* cells.
https://elifesciences.org/articles/60419#fig1video3

**Figure 1—video 4.** Spinning disk confocal microscopy video of Fim1-GFP in *dip1Δ wsp1ΔCA Schizosaccharomyces pombe* cells.
https://elifesciences.org/articles/60419#fig1video4

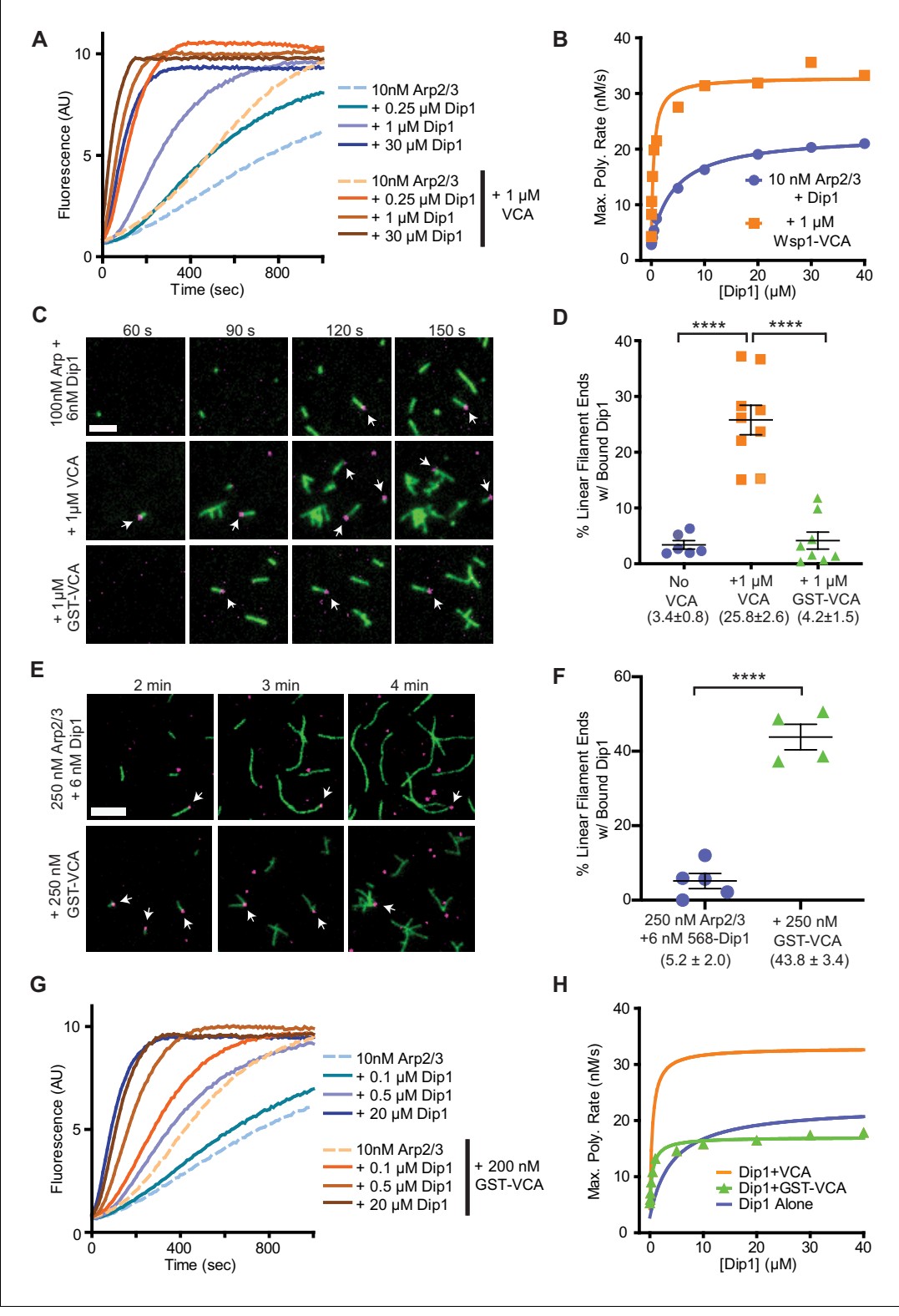

**Figure 2.** Wsp1-VCA increases the number of linear filaments nucleated by Dip1-bound Arp2/3 complex. (**A**) Time courses of polymerization of 3 μM 15% pyrene-labeled rabbit skeletal muscle actin in the presence of 10 nM *Schizosaccharomyces pombe* Arp2/3 complex (SpArp2/3 complex) and 0–30 μM *S. pombe* Dip1 (Dip1) with or without 1 μM *S. pombe* Wsp1-VCA (Wsp1-VCA). (**B**) Plot of the maximum polymerization rates in pyrene-labeled actin polymerization assays as described in A. Data were fit to a hyperbolic equation as described in
*Figure 2 continued on next page*

*Figure 2 continued*

Materials and methods and in **Supplementary file 1**. Only a subset of the reactions included in this titration is shown in the time courses in A. (C) Total internal reflection fluorescence (TIRF) microscopy images of actin polymerization assays containing 100 nM SpArp2/3, 6 nM Alexa Fluor 568-labeled SpDip1 (568-Dip1)(magenta) and 1.5 µM 33% Oregon Green-labeled actin (green) with or without 1 µM SpWsp1-VCA or 1 µM GST-SpWsp1-VCA. The panels are aligned by the reaction times noted above each column. White arrows indicate actin filament pointed ends bound by 568-Dip1. Scale bar: 2 µm. (D) Quantification of the percentage of linear actin filament pointed ends bound by 568-Dip1 2 min and 30 s into actin polymerization assays in C. Branched actin filaments were not included in the calculation. Dip1 binds very weakly to Arp2/3 complex on the ends of preexisting actin filaments under these conditions, so nearly all (~97%) of the pointed ends decorated with Dip1 represent actin filaments nucleated by Dip1-bound Arp2/3 complex (**Balzer et al., 2018**). The total number of linear actin filaments was corrected to account for an approximately twofold decrease in the number of spontaneously nucleated actin filaments caused by inhibition of spontaneous nucleation by GST-Wsp1-VCA or Wsp1-VCA (see **Figure 2—figure supplement 1**). Error bars represent the mean with standard error. p-values: ****<0.0001. (E) TIRF microscopy images of actin polymerization assays containing 250 nM SpArp2/3, 6 nM 568-Dip1 (magenta), and 1.5 µM 33% Oregon Green-labeled actin (green) in the presence or absence of 250 nM GST-SpWsp1-VCA. The panels are aligned by the reaction times noted above each column. White arrows indicate actin filament pointed ends bound by 568-Dip1. Scale bar: 3 µm. (F) Quantification of the percentage of linear actin filament pointed ends bound by 568-Dip1 2 min and 30 s into actin polymerization assays in E. Calculation of the percent bound was carried out as in D. Error bars represent the mean with standard error. p-values: ****<0.0001. (G) Time courses of polymerization of 3 µM 15% pyrene-labeled actin in the presence of 10 nM SpArp2/3 complex and 0–20 µM Dip1 with or without 200 nM GST-SpWsp1-VCA. (H) Plot of the maximum polymerization rate in pyrene-labeled actin polymerization assays containing GST-Wsp1-VCA and Dip1 as described in G. Only a subset of the reactions included in this titration is shown in the time courses in A. The fits for reactions without Wsp1 or with Wsp1-VCA (panel B) are shown for comparison. See **Supplementary file 1** for details on parameters of fits.

The online version of this article includes the following video, source data, and figure supplement(s) for figure 2:

**Source data 1.** Analysis of TIRF microscopy videos source data.
**Figure supplement 1.** Quantification of raw number of Dip1 bound and spontaneously nucleated actin filaments.
**Figure 2—video 1.** Total internal reflection fluorescence (TIRF) microscopy video of actin polymerization assays containing 100 nM SpArp2/3, 6 nM Alexa Fluor 568-labeled SpDip1 (568-Dip1)(magenta), and 1.5 µM 33% Oregon Green-labeled actin (green).
https://elifesciences.org/articles/60419#fig2video1

## Wsp1 synergizes with Dip1 and Arp2/3 complex to produce linear actin filaments

Our bulk actin polymerization assays demonstrate that Wsp1 and Dip1 synergize to activate Arp2/3 complex, but it is unclear whether synergetic activation requires that the complex bind a preformed actin filament, as required when Wsp1 activates Arp2/3 complex on its own. Therefore, it is unclear whether the synergistic activation mechanism could explain how Wsp1 contributes to initiation of new endocytic actin patches. To better understand how the two NPFs synergize, we used single molecule TIRF microscopy to monitor the assembly of Oregon Green 488-labeled actin in the presence of Arp2/3 complex and Wsp1, Dip1, or both NPFs. We labeled Dip1 with Alexa Fluor 568 (568-Dip1) to mark actin filaments nucleated by Arp2/3 complex and Dip1. In the presence of Arp2/3 complex and 568-Dip1, we observed assembly of linear filaments, a subset of which had Dip1 bound at one end (**Figure 2C**). These filaments, which largely represent Dip1-Arp2/3 nucleated filaments (**Balzer et al., 2018**), account for 3.4% of the total filaments in the reaction after two-and-a-half minutes, with the other 96.7% arising from spontaneous nucleation (**Figure 2C and D**, **Figure 2—figure supplement 1**). Adding Wsp1-VCA to the reaction significantly increased both the total number and the percentage of linear filaments with bound Dip1 (**Figure 2C and D** – **Figure 2—figure supplement 1**). These data demonstrate that synergistic activation of Arp2/3 complex by the two NPFs results in the nucleation of linear rather than branched actin filaments. Therefore, we conclude that like Dip1-mediated activation (**Wagner et al., 2013**), synergistic co-activation by both NPFs does not require a preexisting actin filament.

Because Wsp1 may function as an oligomer when clustered at endocytic sites (**Padrick et al., 2008**), we also tested if Wsp1-VCA dimerized with glutathione S-transferase (GST) synergizes with Dip1 to activate Arp2/3 complex. Under our initial reaction conditions (100 nM Arp2/3 complex and

1 µM GST-VCA), we did not detect synergistic co-activation of the complex by Dip1 and GST-VCA (*Figure 2D*). However, in reactions in which the concentration of Arp2/3 complex was increased 2.5-fold, the number and percentage of pointed ends with bound Dip1 increased significantly in the presence of GST-VCA, indicating dimeric (GST) Wsp1-VCA synergizes with Dip1 (*Figure 2E and F*, *Figure 2—figure supplement 1*). To further investigate the influence of Wsp1 dimerization on synergy, we compared the influence of monomeric and dimeric Wsp1-VCAs on the maximum polymerization rate in bulk pyrene actin polymerization assays containing Arp2/3 complex and a range of Dip1 concentrations (*Figure 2G*). These data showed that both dimeric and monomeric Wsp1-VCA significantly decrease the concentration of Dip1 required to saturate the reaction (*Figure 2H*, *Supplementary file 1*). However, unlike monomeric Wsp1-VCA, dimeric Wsp1-VCA did not increase the maximum polymerization rate at saturating Dip1. These data demonstrate that monomeric Wsp1 is more potent in its synergy with Dip1 than dimeric Wsp1, and point to differences in the mechanism of synergistic activation between monomeric and dimeric Wsp1-VCA (see Discussion).

## Actin monomers stimulate activation of Arp2/3 complex by Dip1

To determine the mechanism of co-activation by Dip1 and Wsp1, we first examined the kinetics of activation by Dip1 alone to identify steps in the activation pathway that might be accelerated by Wsp1. We measured time courses of actin polymerization in reactions containing actin, *S. pombe* Arp2/3 complex, and a range of concentrations of Dip1 (0–15 µM) and asked whether various kinetic models were consistent with the polymerization time courses (*Figure 3A*). In the simplest model we considered (*Figure 3B* [model i]), Dip1 binds to Arp2/3 complex and initiates an irreversible activation step to create a filament barbed end. This step could represent an activating conformational change, such as movement of Arp2 and Arp3 into the short pitch helical arrangement or subunit flattening of Arp3 (*Figure 3B*; *Rodnick-Smith et al., 2016b*; *Shaaban et al., 2020*; *Wagner et al., 2013*). The value of the irreversible activation step was floated in the simulations, and the other rate constants were fixed or restrained as described in the supplementary materials. This simple model produced simulated time polymerization courses that fit the data poorly compared to the measured time courses (*Figure 3A*, *Figure 3—figure supplements 1* and *2*). Specifically, the simulations predicted faster polymerization than observed at time points near steady state when the concentration of free actin monomers is low. Therefore, we wondered whether collision with and binding of one or more actin monomers to the Dip1-Arp2/3 assembly might be required to complete the activation process and create a nucleus. To test this, we altered the kinetic model to include one or more actin monomer-binding steps before creation of the nucleus (*Figure 3B*, [models ii and iii]). These models produced simulated polymerization time courses that fit the data significantly better than the reaction pathway without actin monomer collisions (*Figure 3C and D*, *Figure 3—figure supplements 1* and *2*). The pathway with one actin monomer-binding step fits the data most closely, but the fits with two or three monomer-binding steps also improved the fit over the reaction pathway without actin monomer binding (*Figure 3D*). These data suggest that actin monomer binding to the Dip1-bound Arp2/3 complex stimulates activation.

## Actin monomer recruitment by Wsp1 is required for co-activation of Arp2/3 complex by Wsp1 and Dip1

Our data suggest that slow binding of actin monomers to Dip1-bound Arp2/3 complex limits the nucleation rate. Importantly, unlike Dip1, Wsp1 binds both Arp2/3 complex and actin monomers, so it can directly recruit actin monomers to nascent nucleation sites (*Beltzner and Pollard, 2008*; *Marchand et al., 2001*). We wondered if Wsp1 synergizes with Dip1 by recruiting actin monomers to the Dip1-Arp2/3 complex assembly. To test this, we asked whether the actin monomer-recruiting V region of Wsp1 is required for synergistic co-activation of Arp2/3 complex by Dip1 and Wsp1. We found that while adding Wsp1-VCA to actin polymerization reactions containing saturating Dip1 increased the maximal polymerization rate ~1.6-fold compared to reactions without Wsp1, addition of Wsp1-CA had little or no effect on the maximum polymerization rate (*Figure 3E and F*). Therefore, we conclude that actin monomer recruitment by Wsp1 is required for potent synergy with Dip1. Wsp1-CA slightly decreased the concentration of Dip1 required for half maximal saturation ($K_{1/2}$), suggesting that it influences one or more of the activation steps (*Supplementary file 1*).

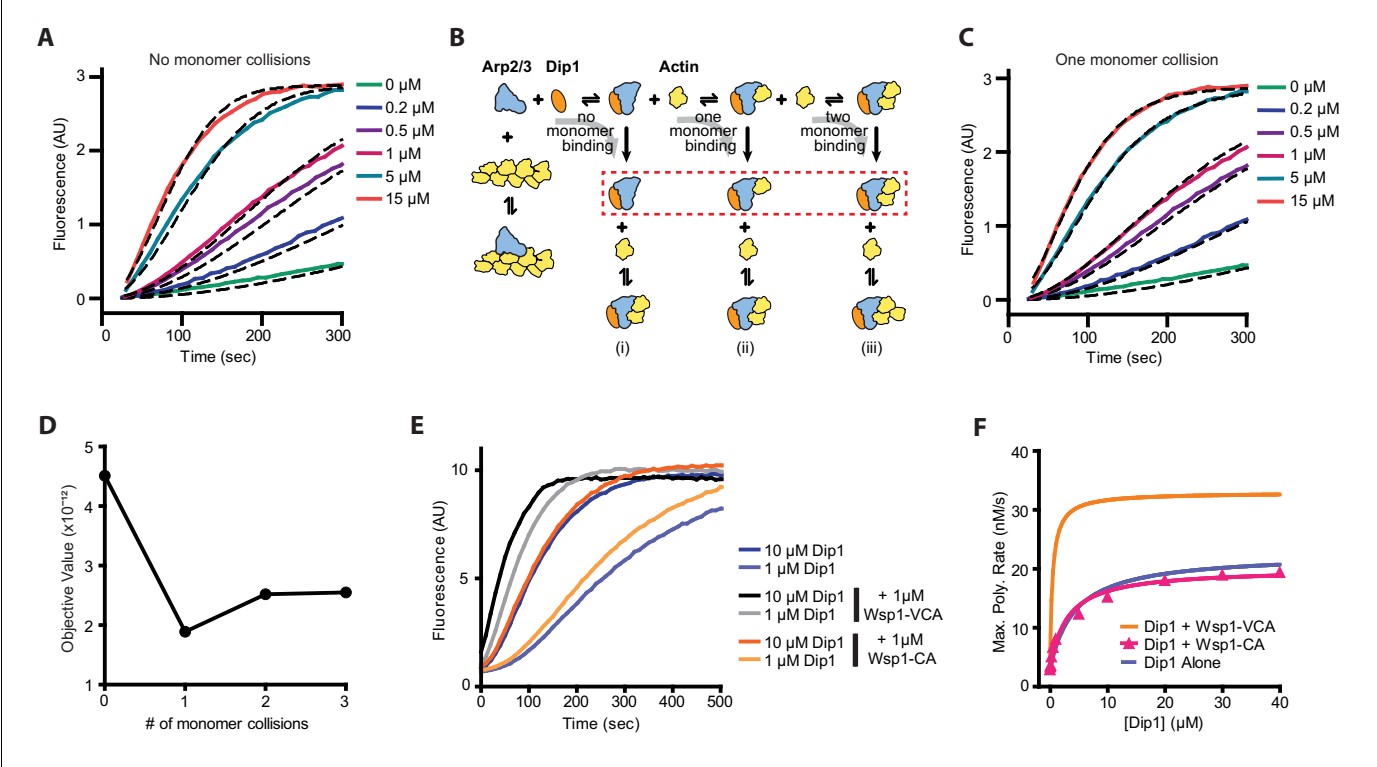

**Figure 3.** Monomer recruitment by Wsp1-VCA is required for maximal co-activation of Arp2/3 complex with Dip1. (**A**) Plot of time courses of polymerization of 3 µM 15% pyrene-labeled actin in the presence of 50 nM SpArp2/3 complex and 0–15 µM Dip1 (solid colored lines). Dashed lines over each trace show the best fits from the no monomer collision model in B. Only a subset of the reactions and fits used in the simulation is shown. For the complete dataset see *Figure 3—figure supplement 1*. (**B**) Schematic showing kinetic pathways used to fit the experimental polymerization time courses. Dashed red lines indicate the nucleus in each of the three pathways tested. For additional details see *Figure 3—figure supplements 1* and *2* and *Supplementary file 2*. (**C**) Plot of time courses of pyrene actin polymerization (solid lines) as in A, with dashed lines over each trace indicating the best fits from the one monomer binding model in B. (**D**) Plot of the objective value obtained from the fits of the pyrene-labeled actin polymerization data in A and C with models containing 0–3 monomer binding events before formation of the nucleus. The objective value represents the normalized mean square weighted sum of squares (see Materials and methods). (**E**) Time courses of polymerization of 3 µM 15% pyrene-labeled actin containing 10 nM SpArp2/3 complex and 1 µM or 10 µM Dip1 with or without 1 µM SpWsp1-CA or 1 µM SpWsp1-VCA. (**F**) Plot of the maximum polymerization rates of pyrene-labeled actin polymerization assays as shown in E. The fits for reactions without Wsp1 or with Wsp1-VCA (*Figure 2B*) are shown for comparison. Data points were fit as described in the Materials and methods. See *Supplementary file 1* for details on parameters of fits.

The online version of this article includes the following source data and figure supplement(s) for figure 3:

**Source data 1.** Source data for pyrene actin polymerization time courses and simulations.
**Figure supplement 1.** Simulated and measured time courses of actin polymerization reactions containing Dip1 and Arp2/3 complex.
**Figure supplement 2.** Full models used to fit actin polymerization time courses in reactions containing Dip1 and Arp2/3 complex with sensitivity analysis of floated parameters.

However, this influence was small compared to the reduction in the $K_{1/2}$ of Dip1 caused by Wsp1 with a V region (*Supplementary file 1*).

## Increased monomer affinity for the nascent nucleus cannot explain synergy on its own

Our data show that Wsp1 must recruit actin monomers to Arp2/3 complex to potently synergize with Dip1 in activation. To better understand how actin monomer recruitment contributes to synergy we sought to kinetically model synergistic activation of Arp2/3 complex by the two NPFs. To decrease the number of unknown rate constants inherent in an explicit model of all reactions in a mixture containing Dip1, Wsp1, and Arp2/3 complex, we used a simplified model based on the activation by Dip1 alone (Figure 4A and B). We asked if this simplified model could fit time courses of actin polymerization for reactions containing both Dip1 and Wsp1 if the rate constants of key steps

were increased. We limited the fitting to reactions with Dip1 concentrations greater than 0.5 µM, as higher concentrations of Dip1 limit the contribution of branching nucleation to actin assembly (*Balzer et al., 2019*). This allowed us to ignore the action of Wsp1 alone on Arp2/3 complex in the simulated reactions. Given that Wsp1 directly tethers actin monomers to the complex, we first asked whether increasing the monomer affinity for the Dip1-Arp2/3 assembly could explain the increased rate of filament nucleation. To test this, we simulated polymerization using the Dip1 alone activation model and allowed the off rate ($k_{-10}$) for the actin monomer bound to the Dip1-Arp2/3 assembly to float. All other rate constants were fixed at the values determined for reactions without Wsp1. These simulations fit the data poorly, indicating the synergy between Wsp1 and Dip1 cannot be explained by increased affinity of actin monomers for the Dip1-Arp2/3 assembly alone (*Figure 4B and C*). However, when we floated the dissociation constant for actin monomers ($k_{-10}$) and either the $k_{off}$ of Dip1 for Arp2/3 complex ($k_{-9}$) or the rate constant for the activation step ($k_{11}$) (or all three), the simulations closely matched the measured polymerization time courses (*Figure 4C–E*, *Supplementary file 3*). These observations suggest that multiple steps in the Dip1-mediated activation pathway are accelerated when Wsp1-bound actin monomers are recruited to Arp2/3 complex. We note that a similar set of simulations in which the association rate constants were floated for the actin monomer or Dip1-binding steps (instead of the dissociation rate constants) yielded the same results (*Figure 4— figure supplement 1*).

We wondered whether the activating step(s) represented by $k_{11}$ might be one of the steps accelerated in the presence of both Dip1 and Wsp1. This step could represent activating conformational

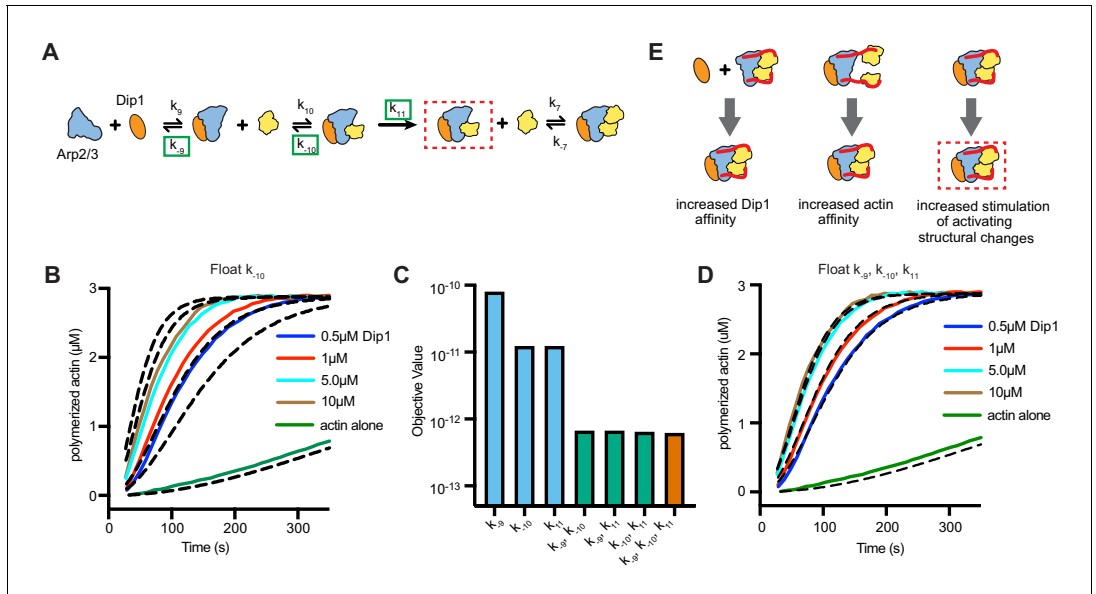

**Figure 4.** Wsp1-bound monomer recruitment accelerates multiple steps of the Dip1-mediated activation pathway. (**A**) Simplified kinetic model of synergistic activation of Arp2/3 complex by Dip1 and Wsp1 based on the Dip1 alone 'one monomer-binding' activation pathway from *Figure 3B*. Note that Wsp1-VCA is not explicitly included in the model. Rate constants boxed in green were floated to fit time courses of reactions that contained both Dip1 and monomeric Wsp1-VCA. The purpose of this simplified model is to test the potential influence of Wsp1-mediated actin monomer recruitment on the steps of Dip1-mediated activation of Arp2/3 complex highlighted in E. (**B**) Plot of time courses of polymerization of 3 µM 15% pyrene-labeled actin in the presence of 50 nM SpArp2/3 complex, 1 µM Wsp1-VCA, and a range of Dip1 from 0.5–10 µM (solid colored lines). Dashed lines over each trace indicate the best fits from the model where only the off rate constant of the actin monomer bound to Dip1-Arp2/3 complex ($k_{-10}$) was floated. (**C**) Objective values obtained from models floating the indicated parameters. The objective value represents the normalized mean square weighted sum of squares (see Materials and methods). (**D**) Plot of time courses shown in B with dashed lines over each trace indicating the best fits from a model in which $k_{-9}$, $k_{-10}$, and $k_{11}$ were floated. (**E**) Depiction of the steps in Dip1-mediated activation of Arp2/3 complex that may be influenced by monomer recruitment. Dashed red lines in A and E indicate the nucleation competent state.

The online version of this article includes the following source data and figure supplement(s) for figure 4:

**Source data 1.** Source data for pyrene actin polymerization time courses and simulations in *Figure 4*.

**Figure supplement 1.** Floating the association rate constants $k_9$ and $k_{10}$ in kinetic simulations of synergistic activation by Dip1 and Wsp1.

**Figure supplement 2.** Additional experimental methods to determine how Wsp1 accelerates Dip1-mediated activation of Arp2/3 complex.

changes that occur in the recently published cryoEM structure of activated Arp2/3 complex bound to Dip1 (*Shaaban et al., 2020*). During activation, approximately one-half of the complex rotates into a new position, moving Arp2 and Arp3 into a conformation (the short pitch conformation) that mimics an actin dimer across the short pitch helix of an actin filament. In addition, each actin-related protein undergoes an intrasubunit conformational change called flattening. While no available assays probe flattening, we took advantage of a previously developed engineered site-specific crosslinking assay to test the influence of Wsp1 and Dip1 on stimulation of the short pitch conformation (*Hetrick et al., 2013*; *Rodnick-Smith et al., 2016a*). In this assay, the crosslinker bis-maleimidoethane (BMOE) (8 Å spacer length) crosslinks a pair of engineered cysteine residues only when Arp2 and Arp3 are in or near a short pitch arrangement. As expected, Wsp1 and Dip1 alone each stimulate short pitch crosslinking (*Figure 4—figure supplement 2*). Addition of a saturating concentration of Dip1 to a saturating concentration of Wsp1 increased short pitch crosslinking over that observed for either NPF alone, suggesting the two NPFs together may accelerate this activating step during synergistic activation. Dip1 also increased short pitch crosslinking when added to a reaction containing Wsp1 and actin monomers.

Given that co-stimulation of the short pitch conformation by Wsp1 and Dip1 would require simultaneous binding of Arp2/3 complex by the two NPFs, we also asked if Wsp1 and Dip1 co-bind to Arp2/3 complex. GST-tagged Wsp1-VCA was able to pull down a significant amount of Dip1 in the presence but not the absence of Arp2/3 complex, indicating formation of a Wsp1-Arp2/3-Dip1 assembly (*Figure 4—figure supplement 2*). Detection of the Dip1-Arp2/3-Wsp1 assembly with this method required using a mutant Arp2/3 complex that binds with increased affinity to Wsp1, presumably because the Dip1-Arp2/3 interaction is weak (*Wagner et al., 2013*).

## Discussion

Here we propose a model in which Wsp1 synergizes with Dip1 to activate Arp2/3 complex and initiate the assembly of endocytic actin networks. Previous measurements of the dynamics of fluorescently labeled NPFs support this model, because they show that the two NPFs colocalize at endocytic sites and arrive with nearly identical timing, ~2 s before actin filaments begin to polymerize (*Basu and Chang, 2011*; *Sirotkin et al., 2010*). Given that Dip1 is biochemically specialized to initiate branched actin network assembly, it might be expected to peak in concentration before actin begins to polymerize. However, the accumulation kinetics of Dip1 are nearly identical to Wsp1; both accumulate over ~6 s as actin assembles, reach a peak concentration just before (~2 s) the actin filament concentration peaks, and then dissociate as the patch begins to internalize and actin disassembles (*Basu and Chang, 2011*). The gradual accumulation of Dip1 suggests that it might co-activate Arp2/3 complex with Wsp1 well after actin polymerization has been initiated and throughout the assembly/propagation of the actin patch. This activity would have implications for determining the architecture of actin filament networks at the endocytic sites (see below).

Relatively little is known about how the NPFs that control actin assembly at endocytic sites are regulated, despite their importance in driving endocytosis. In *Saccharomyces cerevisiae*, both inhibitors and activators of the WASP family protein Las17 have been identified (*Goode et al., 2015*), and recent experiments suggest that clustering Las17 at high concentrations at endocytic sites might trigger its activity (*Sun et al., 2017*). Building evidence suggests that homologues of the proteins that regulate Las17 in *S. cerevisiae* may control Wsp1 activity in *S. pombe*

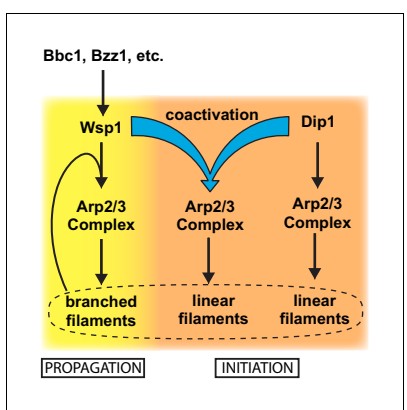

**Figure 5.** Dip1 and Wsp1 regulation may be coordinated through co-activation of Arp2/3 complex. Schematic of the activation pathways of Arp2/3 complex by Dip1 and Wsp1. Nucleation of linear actin filaments by Arp2/3 complex is critical for initiation of actin networks, while branching nucleation promotes propagation of actin networks. Regulatory factors required for Wsp1 activation can indirectly trigger Dip1 activity by stimulating co-activation of Arp2/3 complex by both nucleation promoting factors.

(*Arasada and Pollard, 2011*; *MacQuarrie et al., 2019*), perhaps at least partially by clustering it at endocytic sites. Almost nothing is known about the regulation of Dip1, but we show here that Wsp1 synergizes with Dip1, so activators that turn on Wsp1 could also indirectly stimulate Dip1 (*Figure 5*). Therefore, an attractive hypothesis is that the activation pathway for Wsp1 stimulates both Wsp1 and Dip1, thereby coordinating the initiation and propagation phases of endocytic actin assembly (*Figure 5*). However, some evidence supports the existence of a distinct activation pathway for Dip1. For instance, the N-terminal ~160 amino acids of Dip1 is not required for activity (*Wagner et al., 2013*), but this segment is relatively well conserved among yeast, so it may play a role in localizing or regulating Dip1 independent of Wsp1. Given that initiation is a key step in regulating the assembly of branched actin networks, elucidating how cells control the activity of Dip1 will be an important future goal.

Electron microscopy studies indicate that endocytic actin networks are branched (*Young et al., 2004*), and the highly dendritic nature of these filamentous networks is thought to allow them to drive invagination of the plasma membrane (*Lacy et al., 2018*). Wsp1 creates a branched actin filament when it activates Arp2/3 complex on its own, but we show here that when Wsp1 activates the complex with Dip1 it creates a linear actin filament. This observation suggests that cells may need to limit synergistic activation of Arp2/3 complex by Dip1 and Wsp1 to preserve the dendritic nature of endocytic actin networks. We anticipate that synergistic activation by Wsp1 and Dip1 is limited by the same mechanisms that prevent Dip1 alone from activating too many Arp2/3 complexes at endocytic sites. For instance, we showed previously that when Dip1 activates on its own, it remains bound to Arp2/3 complex long after nucleation, unlike Wsp1, so each Dip1 molecule likely only activates one Arp2/3 complex (*Balzer et al., 2019*). We found here that even when it activates with Wsp1, Dip1 stays bound to Arp2/3 complex on the ends of filaments long after nucleation, so Dip1 likely functions as a single turnover NPF in the context of synergistic activation (*Figure 2*). Combined with the low concentration of Dip1 at endocytic actin patches, this single turnover mechanism may help limit the number of linear filaments created at endocytic sites (*Balzer et al., 2019*; *Basu and Chang, 2011*). Competition with actin filaments may provide a second mechanism for limiting linear filaments generated through synergy between Wsp1 and Dip1. We showed previously that actin filaments compete with WDS proteins for binding to Arp2/3 complex (*Luan et al., 2018a*). Therefore, even if both NPFs are present, activation by Wsp1 alone may dominate once actin filaments begin to accumulate at endocytic sites.

Our simulations of actin polymerization kinetics indicate that when Dip1 activates Arp2/3 complex on its own, actin monomers collide with and bind to the Dip1-Arp2/3 complex assembly to help create the nucleation-competent state. While the function of bound actin monomers is uncertain, our crosslinking data indicate that actin monomer binding may help stimulate the short pitch conformational change in the Dip1-Arp2/3 complex assembly. The cryo EM structure of activated Arp2/3 complex with Dip1 bound also reveals an intra-subunit conformational change that might be stimulated by binding of actin monomers to the Dip1-Arp2/3 complex assembly (*Shaaban et al., 2020*). This structural change, called flattening, brings Arp2 and Arp3 closer to the conformation that actin subunits adopt in filaments, allowing them to mimic an actin dimer. Importantly, we note that a recent study on the human WDS protein, SPIN90, shows that interaction of SPIN90 with the formin mDia1 increases the ability of SPIN90 to trigger nucleation by Arp2/3 complex (*Cao et al., 2020*). Synergy between mDia1 and SPIN90 may occur because mDia1 can recruit actin monomers to the nascent nucleus, using a mechanism analogous to the mechanism of synergy between Dip1 and Wsp1 we propose here (*Cao et al., 2020*).

We show here that direct tethering of actin monomers by monomeric Wsp1 potently accelerates activation of Arp2/3 complex by Dip1, allowing the two NPFs to synergize. Direct tethering of actin monomers to the Dip1-Arp2/3 complex assembly increases their effective concentration, which could potentially explain synergy between Dip1 and Wsp. However, a kinetic model that accounted for the increased effective concentration (by decreasing the off rate of actin monomers for the Dip1-Arp2/3 assembly) could not fully explain the acceleration of actin polymerization in reactions containing both NPFs (*Figure 4*). To accurately simulate synergistic activation, our models allowed Wsp1-recruited actin to either increase the affinity of Dip1 for Arp2/3 complex or accelerate the final activation step (*Figure 4*). While it is unclear whether one or both of these additional steps are influenced during synergistic activation, our crosslinking data suggest that Wsp1-recruited actin monomers may stimulate the activating step ($k_{11}$ in our simulations) more rapidly than randomly

colliding and binding actin monomers. Understanding the molecular basis for the acceleration of each affected step during synergistic activation will be important for understanding how Dip1 and Wsp1 activate Arp2/3 complex together and on their own.

A surprising result of this work is that Wsp1 dimerized by GST showed significantly less synergy with Dip1 than monomeric Wsp1. While dimerized Wsp1 moderately decreased the amount of Dip1 required to reach half maximal saturation ($K_{1/2}$, see *Supplementary file 1*), at saturating Dip1, the maximum polymerization rate was less with GST-Wsp1-VCA than without it. Given that dimeric WASP proteins can recruit two actin monomers and typically bind ~100–150-fold more tightly to the complex than monomeric WASP proteins (*Padrick et al., 2011*; *Padrick et al., 2008*), we initially expected that dimeric Wsp1 would have greater synergy with Dip1 than Wsp1 monomers. However, previous biochemical and structural data indicate that WASP proteins – when activating on their own – must be released from nascent branch junctions before nucleation (*Helgeson and Nolen, 2013*; *Smith et al., 2013b*). Because they likely bind the branch junction more tightly (*Helgeson and Nolen, 2013*), dimeric WASP proteins are thought to release more slowly, thereby decreasing how fast nucleation occurs once WASP is bound compared to monomeric WASP. Likewise, tight binding by dimeric Wsp1 to the nascent linear filament nucleus could slow its release, thereby decreasing the nucleation rate and diminishing synergy between Dip1 and Wsp1. The significant differences we observed between dimeric and monomeric Wsp1 in synergizing with Dip1 highlight the need to better understand how Wsp1 activates Arp2/3 complex in cells. Recent experiments in budding yeast showed that Las17 is recruited to endocytic sites through a set of multivalent interactions similar to the types of interactions that incorporate WASP proteins into phase separated droplets in vitro (*Banjade and Rosen, 2014*; *Li et al., 2012*; *Sun et al., 2017*). Whether or not Wsp1 accumulates in similar phase separated droplets, it will be important to understand if Wsp1 engages Arp2/3 complex as a monomer or oligomer, as this will significantly influence the kinetics of its activation of the complex alone or with Dip1.

# Materials and methods

## Key resources table

| Reagent type (species) or resource | Designation | Source or reference | Identifiers | Additional information |
|---|---|---|---|---|
| Antibody | Anti-Dip1 (rabbit polyclonal) | GenScript (custom) | n/a | WB (1:5000) |
| Antibody | Anti-ScArp3 (goat polyclonal) | Santa Cruz | Cat# sc-11973 | WB (1:1000) |

### Protein expression, purification, and fluorescent labeling

To purify *S. pombe* Dip1, an N-terminally GST tagged Dip1 plasmid was generated by cloning the full length Dip1 sequence into the pGV67 vector as described previously (*Balzer et al., 2018*). The restriction sites chosen for cloning resulted in the presence of a short N-terminal polypeptide sequence (GSMEFELRRQACGR) on the end of the coding sequence for Dip1 after cleavage with tobacco etch virus (TEV). To purify Dip1, BL21(DE3) RIL *Escherichia coli* cells transformed with this pGV67-Dip1 plasmid were grown to an O.D. 595 of 0.6–0.7, induced with 0.4 mM isopropyl 1-thio-β-D-galactopyranoside (IPTG), and incubated overnight at 22˚C. Cells were lysed by sonication in lysis buffer (20 mM Tris pH 8.0, 140 mM NaCl, 2 mM EDTA, 1 mM dithiothreitol [DTT], 0.5 mM phenylmethylsulfonyl fluoride [PMSF], and protease inhibitor tablets [Roche]) and then clarified by centrifugation (JA-20 rotor [Beckman], 18,000 rpm, 25 min, 4˚C). The supernatant was pooled and loaded onto a 10 mL glutathione sepharose 4B (GS4B) column equilibrated in GST-binding buffer (20 mM Tris pH 8.0, 140 mM NaCl, 2 mM EDTA, 1 mM DTT). The column was then washed with GST-binding buffer until no protein was detected in the flow through (~10 CV). Protein was eluted with 20 mM Tris pH 8.0, 140 mM NaCl, and 50 mM glutathione. Fractions containing GST-Dip1 were pooled and dialyzed overnight against 20 mM Tris pH 8.0, 50 mM NaCl, and 1 mM DTT at 4˚C in the presence of TEV protease (25:1 ratio of GST-Dip1 to TEV protease [by mass]). The dialysate was loaded onto a 6 mL Resource Q column equilibrated in $Q_A$ buffer (20 mM Tris pH 8.0, 50 mM NaCl, and 1 mM DTT) and eluted over a 20 column volume gradient to 100% $Q_B$ buffer (20 mM Tris pH 8.0, 500 mM

NaCl, and 1 mM DTT). The protein was concentrated in a 10 k MWCO Amicon-Ultra centrifugal filter (Millipore Sigma) and loaded onto a Superdex 200 HiLoad 16/60 gel filtration column equilibrated in 20 mM Tris pH 8.0, 100 mM NaCl, and 1 mM DTT. Fractions containing pure Dip1 were pooled and flash frozen in liquid nitrogen. The final concentration of *S. pombe* Dip1 was determined by the absorbance at 280 nm using an extinction coefficient of 36,330 $M^{-1}cm^{-1}$.

The Dip1 construct used for site-specific labeling with the cysteine-reactive Alexa Fluor 568 C5 maleimide (Thermo Fisher) had the six endogenous cysteine residues in Dip1 mutated to alanine and a single cysteine added to the short N-terminal polypeptide sequence on the end of the coding sequence as previously described (*Balzer et al., 2018*). Expression and purification of this Dip1 mutant were identical to the wild-type purification until the protein was loaded onto the Superdex 200 HiLoad 16/60 gel filtration column. To increase labeling efficiency, the size exclusion column was equilibrated in 20 mM HEPES pH 7.0 and 50 mM NaCl before loading and eluting the concentrated protein. Peak fractions containing Dip1 were pooled and concentrated to 40 µM for labeling. A 10 mM solution of Alexa Fluor 568 C5 maleimide dye in water was added dropwise to the protein while stirring at 4°C until the solution reached a 10−40 molar excess of dye to protein. After 12–16 hr the reaction was quenched by dialyzing against 20 mM Tris pH 8.0, 50 mM NaCl, and 1 mM DTT for 24 hr. Labeled Dip1 was loaded onto a Hi-Trap desalting column and the peak fractions were pooled and flash frozen in liquid nitrogen. The final concentration of Alexa Fluor 568 dye was determined by measuring the absorbance at 575 nm and dividing this by 92,009 $M^{-1}cm^{-1}$, the extinction coefficient of the dye. The final concentration of 568-Dip1 was determined by the following equation: $\frac{A_{280} - (A_{575} \times 0.403)}{36,330 M^{-1}cm^{-1}}$.

To purify *S. pombe* Wsp1-VCA, residues 497–574 were cloned into the pGV67 vector containing an N-terminal GST tag followed by a TEV cleavage site. A 5 mL culture of BL21(DE3)-RIL *E. coli* cells transformed with the pGV67-Wsp1-VCA vector in LB plus 100 µg/mL ampicillin and 35 µg/mL chloramphenicol was grown overnight at 37°C. One milliliter of this culture was used to inoculate 50 mL of LB plus ampicillin and chloramphenicol, which was allowed to grow at 37°C with shaking until turbid. Ten milliliters of this turbid culture were added to a 2.8 L flask containing 1 L of LB plus ampicillin and chloramphenicol and grown to an O.D. 600 of 0.4–0.6 before inducing by adding IPTG to 0.4 mM. Cells were allowed to express for 12–14 hr at 22°C before adding EDTA and PMSF to 2 mM and 0.5 mM, respectively. Cells were then harvested and lysed by sonication in lysis buffer (20 mM Tris pH 8.0, 140 mM NaCl, 2 mM EDTA, 1 mM DTT, 0.5 mM PMSF, and protease inhibitor tablets [Roche]) and then clarified by centrifugation (JA-20 rotor [Beckman], 18,000 rpm, 25 min, 4°C). The clarified lysate was then loaded onto a 10 mL GS4B column equilibrated in GST-binding buffer (20 mM Tris pH 8.0, 140 mM NaCl, 2 mM EDTA, and 1 mM DTT) and washed with ~10 column volumes of the same buffer. Protein was eluted with approximately three column volumes of elution buffer (20 mM Tris pH 8.0, 100 mM NaCl, 1 mM DTT, and 50 mM reduced L-glutathione) and fractions containing GST-SpWsp1-VCA were pooled and dialyzed overnight in 3500 MWCO tubing against 2 L of 20 mM Tris pH 8.0, 50 mM NaCl, and 1 mM DTT at 4°C in the presence of a 1:25 ratio (by mass) of TEV protease to recombinant protein. To purify the GST tagged Wsp1-VCA, the addition of TEV to the dialysis was omitted. The dialysate was loaded onto a 6 mL Source30Q column equilibrated in $Q_A$ buffer (20 mM Tris pH 8.0, 50 mM NaCl, and 1 mM DTT) and eluted over a 20 column volume gradient to 100% $Q_B$ buffer (20 mM Tris pH 8.0, 500 mM NaCl, and 1 mM DTT). Fractions containing GST-Wsp1-VCA were concentrated to 1.5 mL and loaded onto a Superdex 75 gel filtration column equilibrated in 20 mM Tris pH 8.0, 150 mM NaCl, and 1 mM DTT. Fractions containing pure protein were pooled and concentrated in a 3500 MWCO Amicon-Ultra centrifugal filter (Millipore Sigma) at 4°C. The concentrated, pure protein was flash frozen in liquid nitrogen. The final concentration of *S. pombe* Wsp1-VCA was determined by measuring the absorbance at 280 nm and dividing this by the Wsp1 extinction coefficient of 5500 $M^{-1}cm^{-1}$.

To construct an expression plasmid for *S. pombe* Wsp1-CA, residues 519–574 were cloned into the pGV67 vector containing an N-terminal GST tag followed by a TEV cleavage site. The purification was carried out as described for *S. pombe* Wsp1-VCA above.

To purify *S. pombe* Arp2/3 complex, 10 mL of a turbid culture of *S. pombe* (strain TP150) cells was added to a 2.8 L flask containing 1 L of YE5S. Cultures were grown for ~12 hr at 30°C with shaking followed by the addition of solid YE5S media, equivalent to the mass required for 1 L of liquid culture, and growth for ~4 more hours. Before harvesting, EDTA was added to a final concentration

of 2 mM. All subsequent steps were carried out at 4°C. The cultures were centrifuged, and the pellet was resuspended in 2 mL of lysis buffer (20 mM Tris pH 8.0, 50 mM NaCl, 1 mM EDTA, and 1 mM DTT) per gram of wet cell pellet, plus six protease inhibitor tablets (Roche) per liter of lysis buffer. The resuspended cells were lysed in a microfluidizer (Microfluidics Model M-110EH-30 Microfluidizer Processor) at 23 kPSI over five to six passes. After lysis, PMSF was added to 0.5 mM and the lysate was spun down in a JA-10 (Beckman) rotor at 9000 rpm for 25 min. The supernatant was transferred to prechilled 70 mL polycarbonate centrifuge tubes and spun at 34,000 rpm for 75 min in a Fiberlite F37L rotor (Thermo-Scientific). The pellet was discarded and the supernatant was filtered through cheesecloth into a prechilled graduated cylinder to determine the volume. Under heavy stirring, 0.243 g of ammonium sulfate per milliliter of supernatant was added over approximately 30 min. The solution stirred for an additional 30 min, and then was pelleted in a Fiberlite F37L rotor at 34,000 rpm for 90 min. The pellet was resuspended in 50 mL of PKME (25 mM PIPES, 50 mM KCl, 1 mM EGTA, 3 mM MgCl2, 1 mM DTT, and 0.1 mM ATP) and dialyzed overnight in 50,000 MWCO dialysis tubing against 8 L PKME. The dialysate was clarified by centrifugation in the Fiberlite F37L rotor at 34,000 rpm for 90 min. A 10 mL column of GS4B beads was equilibrated in GST-binding buffer (20 mM Tris pH 8.0, 140 mM NaCl, 1 mM EDTA, and 1 mM DTT) before it was charged with 15 mg of GST-N-WASP-VCA to make a GST-VCA affinity column. The charged column was washed with additional binding buffer until no protein was detectable in the flow through by Bradford assay. The column was then equilibrated in PKME pH 7.0, the supernatant was loaded at 1 mL/min, and the column was washed with additional PKME (~45 mL). A second wash with PKME + 150 mM KCl was done until no protein was detected in the flow through by Bradford assay (~30 mL). Protein was eluted with PKME + 1 M NaCl into ~2 mL fractions until no protein was detected by a Bradford assay (~30 mL). Fractions containing Arp2/3 complex were pooled and dialyzed overnight in 50,000 MWCO dialysis tubing against 2 L of $Q_A$ buffer (10 mM PIPES, 25 mM NaCl, 0.25 mM EGTA, 0.25 mM MgCl2, and pH 6.8 with KOH). Arp2/3 complex was further purified by ion exchange chromatography on an FPLC using a 1 mL MonoQ column with a linear gradient of $Q_A$ buffer to 100% $Q_B$ buffer (10 mM PIPES, 500 mM NaCl, 0.25 mM EGTA, 0.25 mM MgCl2, and pH 6.8 with KOH) over 40 column volumes. Fractions containing Arp2/3 complex were pooled and dialyzed overnight in 50,000 MWCO dialysis tubing against Tris pH 8.0, 50 mM NaCl, and 1 mM DTT. The dialysate was concentrated to 1.5 mL in a 30,000 MWCO concentrator tube (Sartorius Vivaspin Turbo 15 #VS15T21) using the Fiberlite F13B rotor at 2500 rpm over several 5–10 min cycles. Between each cycle the solution was mixed by gentle pipetting. The concentrated sample was loaded on a Superdex 200 HiLoad 16/60 gel filtration column equilibrated in Tris pH 8.0, 50 mM NaCl, and 1 mM DTT. Fractions containing pure Arp2/3 complex were concentrated as described above and the final concentration was determined by measuring the absorbance at 290 nm and dividing by 139,030 $M^{-1}cm^{-1}$, the extinction coefficient ($\varepsilon_{290}$) of Arp2/3 complex, before flash freezing.

Biotin-inactivated myosin was prepared by reacting 2 mg of rabbit skeletal muscle myosin (Cytoskeleton Cat # MYO2) with 5 μL of 250 mM EZ-Link-Maleimide-PEG11-Biotin dissolved in DMSO. The labeling reaction was carried out in 500 μL reaction buffer (20 mM HEPES pH 8.0, 500 mM KCl, 5 mM EDTA, 1 μM ATP, and 1 mM MgCl2) on ice for 6 hr. The biotin–myosin was then dialyzed against 0.5 L of storage buffer (20 mM imidazole pH 7.0, 500 mM KCl, 5 mM EDTA, 1 mM DTT, and 50% glycerol) using a 3500 MWCO dialysis thimble (Thermofisher Slide-A-Lyzer MINI dialysis unit 0069550). The final volume of biotin–myosin was measured, and the concentration was determined based on 1.85 mg/mL equaling 3.86 μM. Biotin–myosin was stored at −20°C.

Actin was purified using a modified method based on the one described by *Spudich and Watt, 1971*. Five grams of rabbit muscle acetone powder was resuspended in 100 mL G buffer (2 mM Tris pH 8.0, 0.2 mM ATP, 0.5 mM DTT, 0.1 mM CaCl2, and 1 mM sodium azide [NaN3]) and stirred at 4°C for 30 min. All additional steps are carried out at 4°C. Resuspended rabbit muscle was centrifuged for 25 min at 16,000 rpm in a Beckman JA-20 rotor and the supernatant was filtered through glass wool into a prechilled graduated cylinder. The pellet was resuspended in an additional 100 mL G buffer, pelleted as described above, and the supernatant was filtered through glass wool and pooled with the original supernatant. The total volume was measured and the solution was brought to 50 mM KCl and 2 mM MgCl2 by adding 25 μL 2 M KCl and 2 μL 1 M MgCl2 per 1 mL of supernatant with stirring. The solution was allowed to stir for 1 hr to polymerize actin. After 1 hr, 0.056 g KCl per 1 mL of original supernatant volume was added to bring the solution to 0.8 M KCl. The solution was stirred for an additional 30 min to dissociate tropomyosin before it was pelleted in a Beckman 70Ti

rotor at 31,700 rpm for 2 hr. The supernatant was discarded and two-third of the pellet was homogenized in 5 mL of G buffer using a Dounce homogenizer and dialyzed in 1 L of G buffer to begin to depolymerize actin filaments. The remaining one-third of the pellet was homogenized in 4 mL of labeling buffer 25 mM Tris pH 7.5, 100 mM KCl, 0.3 mM ATP, 2 mM MgSO$_4$, and 3 mM sodium azide (NaN$_3$) and dialyzed against 1 L of labeling buffer for at least 4 hr to remove DTT. The concentration of actin was then measured ([actin] = A$_{290}$/0.0266 µM$^{-1}$cm$^{-1}$) and actin was diluted to 23.8 µM (1 mg/mL). A 4–7 molar excess of N-(1-pyrene)Iodoacetimide was added to the actin dropwise with stirring. The labeling reaction proceeded for ~14 hr, covered from light. Pyrene-labeled actin was pelleted using a Beckman 90Ti rotor at 36,200 rpm for 2 hr and then homogenized in 1.5 mL G buffer and dialyzed against 1 L G buffer while protected from light. Both unlabeled and pyrene-labeled actin were dialyzed against G buffer for 1.5–2 days with at least three exchanges of dialysis buffer. Following dialysis, depolymerized actin was centrifuged in a Beckman 90 Ti rotor at 36,200 rpm for 2 hr to pellet out any polymerized actin and the top 80% of the supernatant was the gel filtered using S-300 resin and G buffer. Fractions of 3–4 mL were collected over ~300 mL G buffer and the peak fraction was identified using a Bradford assay. The concentration of unlabeled actin was determined using the following equation: [Actin, µM] = A$_{290}$ × 38.5 µM/OD. The concentration of pyrene-labeled actin was determined using the following equation: [Pyrene-labeled actin, µM] = (A$_{290}$ – (0.127 × A$_{344}$)) × 38.5 µM/OD. The labeling percentage of pyrene-labeled actin was determined by dividing the pyrene-labeled actin concentration by the concentration of pyrene ([pyrene, µM] = A$_{344}$ × 45.5 µM/OD). Actin was stored at 4°C with pyrene-labeled actin covered in foil to protect from light.

Oregon Green-labeled actin was purified as pyrene actin until resuspension in labeling buffer. Actin was resuspended in 2 mL labeling buffer (25 mM imidazole pH 7.5, 100 mM KCl, 0.3 mM ATP, 2 mM MgCl$_2$, and 3 mM sodium azide (NaN$_3$)) and dialyzed against 1 L of labeling buffer for at least 4 hr to remove DTT. The concentration of actin was then measured ([actin] = A$_{290}$/0.0266 µM$^{-1}$cm$^{-1}$) and actin was diluted to 23.8 µM (1 mg/mL). A 10–12 molar excess of Oregon Green 488 Maleimide (Invitrogen O-6034) was added to the actin dropwise with stirring. The labeling reaction proceeded for ~14 hr, covered from light. Oregon Green-labeled actin was centrifuged, homogenized, and gel filtered as with pyrene actin. The concentration of pyrene-labeled actin was determined using the following equation: [Oregon Green-labeled actin, µM] = (A$_{290}$ – (0.16991 × A$_{491}$)) × 38.5 µM/OD. The labeling percentage of Oregon Green-labeled actin was determined by dividing the Oregon Green-labeled actin concentration by the concentration of Oregon Green dye ([Oregon Green 488 Maleimide, µM] = A$_{491}$/0.081 µM$^{-1}$cm$^{-1}$). Oregon Green-labeled actin was stored at 4°C covered in foil to protect from light.

## Crosslinking assays

Crosslinking assays were carried out as previously described (*Rodnick-Smith et al., 2016b*). Briefly, 15 µL reactions were initiated by adding BMOE to 20 µM to make a final solution that contained 1 µM dual cysteine SpArp2/3 complex in 10 mM imidazole pH 7.0, 50 mM KCl, 1 mM EGTA, 1 mM MgCl$_2$, 200 µM ATP plus Wsp1-VCA, Dip1, or latrunculin actin as indicated. Reactions were incubated at room temperature for 45 s before quenching with SDS-PAGE loading buffer with fresh DTT. Reactions were analyzed by western blotting using an anti-Arp3 antibody (1:1000).

## TIRF microscopy slide preparation

TIRF flow chambers were constructed as previously described with slight modifications (*Kuhn and Pollard, 2005*). All following cleaning steps were carried out at room temperature. Coverslips (24 × 60 #1.5) were cleaned in Coplin jars by sonicating in acetone followed by 1 M KOH for 25 min each, with a deionized water rinse between each sonication step. Coverslips were then rinsed twice with methanol and aminosilanized by incubating in a 1% APTES (Sigma), 5% acetic acid in methanol solution for 10 min before sonicating for 5 min, and then incubating for an additional 15 min. Coverslips were then rinsed with two volumes of methanol followed by thorough flushing with deionized water. After air drying, TIRF chambers were created by pressing two pieces of double-sided tape onto a cleaned coverslip with a 0.5 cm wide gap between them. A glass microscope slide was then placed on top of the coverslip and tape perpendicularly to create a cross-shape forming a chamber in the middle with a volume of ~14 µL. Chambers were passivated by flowing in 300 mg/mL methoxy PEG

succinimidyl succinate, MW5000 (JenKem) containing 1–3% biotin-PEG NHS ester, MW5000 (Jen-Kem) dissolved in 0.1 M NaHCO$_3$ pH 8.3 and incubating for 4–5 hr. Excess PEG was washed out with 0.1 M NaHCO$_3$ pH 8.3 before flowing deionized water into chambers for storage. Chambers were stored at 4°C for less than 1 week. Immediately prior to imaging, 1 µM NeutrAvidin (Thermo-Fisher) was added to chambers and incubated for 8 min followed by 8 min with 50–150 nM biotin inactivated myosin (Cytoskeleton, Inc), both prepared in 50 mM Tris pH 7.5, 600 mM NaCl. Chambers were washed two times with 20 mg/mL BSA in 50 mM Tris pH 7.5, 600 mM NaCl followed by two washes with 20 mg/mL BSA in 50 mM Tris pH 7.5, 150 mM NaCl. Chambers were finally preincubated with TIRF buffer (10 mM imidazole pH 7.0, 1 mM MgCl$_2$, 1 mM EGTA, 50 mM KCl, 100 mM DTT, 0.2 mM ATP, 25 mM glucose, 0.5% methylcellulose [400 cP at 2%], 0.02 mg/mL catalase [Sigma], and 0.1 mg/mL glucose oxidase [MP Biomedicals]) after which point they were ready to add reaction mixture.

## Actin polymerization reactions in TIRF chambers

In a typical reaction, 1 µL of 2.5 mM MgCl$_2$ and 10 mM EGTA were mixed with 5 µL of 9 µM 33% Oregon Green-labeled actin and incubated for 2 min. Four microliters of the actin solution were then added to 16 µL of a solution containing 1.25× TIRF buffer and any other proteins. Reactions were imaged on a Nikon TE2000 inverted microscope equipped with a 100 × 1.49 numerical aperture TIRF objective, 50 mW 488 nm and 561 nm Sapphire continuous wave solid state laser lines (Coherent), a dual band TIRF (zt488/561rpc) filter cube (Chroma C143315), and a 1× to 1.5× intermediate magnification module. Images were collected using an 512 × 512 pixel EM-CCD camera (iXon3, Andor). For two color reactions, typical imaging conditions were 50 ms exposures with the 488 nm laser (set to 5 mW) and 100 ms exposures with the 561 nm laser (set to 35 mW) for 1 s intervals. The camera EM gain was set to 200. The concentration of 568-Dip1 was kept in the low nanomolar range in all assays to prevent high backgrounds of nonspecifically adsorbed 568-Dip1 from obscuring Dip-Arp2/3 filament nucleation events.

## Pyrene actin polymerization assays

In a typical reaction, 2 µL of 10× ME buffer (5 mM MgCl$_2$ and 20 mM EGTA) was added to 20 µL of 15% pyrene-labeled actin and incubated for 2 min in 96-well flat bottom black polystyrene assay plates (Corning 3686). To initiate the reaction, 78 µL of buffer containing all other proteins was added to the actin wells using a multichannel pipette. This brought the final buffer concentration in the reaction to 10 mM imidazole pH 7.0, 50 mM KCl, 1 mM EGTA, 1 mM MgCl$_2$, 200 µM ATP, and 1 mM DTT. Polymerization of actin was measured by exciting pyrene actin at 365 nm and monitoring the emission at 407 nm using a TECAN Safire two plate reader. The maximum polymerization rate of pyrene actin polymerization assays was determined by measuring the slope of each curve at each time point and converting from RFU/s to actin (nM)/s by assuming that the actin filament concentration was zero at the minimum fluorescence value and 0.1 µM actin was unpolymerized at the maximum fluorescence. The maximum polymerization rate was plotted for a series of reactions with increasing concentrations of Dip1 and fixed concentrations of all other proteins. For these plots, data points were fit to the following equation: Max poly rate = (max poly rate$_{max}$× [Dip1])/(K$_{1/2}$ + [Dip1]) + y-intercept, where K$_{1/2}$ represents the [Dip1] needed to get half-maximum max polymerization rate and the y-intercept represents the maximum polymerization rate in the absence of Dip1. Note that *Figure 2A* shows only a subset of the assays while the maximum polymerization rates of the entire dataset are shown in *Figure 2B*. See *Supplementary file 1* for details on fits.

## Quantification of the number of Dip1-Arp2/3 nucleated actin filaments

The percentage of Dip1-Arp2/3 complex nucleated actin filaments was determined by counting the number of actin filament pointed ends bound by 568-Dip1 in Oregon Green-labeled actin polymerization assays imaged using TIRF microscopy (*Balzer et al., 2018*). The quantification was performed on a region of interest from the movie frame that corresponded to 2 min and 30 s from the initiation of the reaction. All pointed ends present in the quantified frame were tracked from their initial appearance to ensure accuracy in 568-Dip1 pointed end-binding determination. The number of pointed ends bound by 568-Dip1 was divided by the total number of pointed ends in the region of interest. At least four replicate actin polymerization assays were quantified for each condition. The

statistical significance of the data for datasets of only two conditions was tested by Student's t-test in GraphPad Prism. The statistical significance for datasets of more than two conditions was tested by one-way ANOVA with Tukey's post-hoc test in GraphPad Prism. Both tests were two tailed and the significance values are reported in the figure legends.

## Modeling of actin polymerization assays

All modeling was carried out using the open source software application COmplex PAthway SImulator (COPASI) (*Hoops et al., 2006*). Fluorescence values from time courses of polymerization of 3 μM 15% pyrene-labeled actin in the presence of indicated proteins were converted to .txt files using a custom MatLab script and loaded into COPASI software. The actin filament concentrations were determined by assuming 0.1 μM actin was unpolymerized at equilibrium (*Pollard, 1986*). Optimization of reaction parameters was carried out by simultaneously fitting all traces from a reaction set, using the genetic algorithm method in the parameter estimation module.

Models were built by identifying interactions between the components in polymerization assays to build up a set of reactions to describe the polymerization. For many parameters included in our set of reactions, we were able to use previously measured rate constants (see *Supplementary file 2*). Rate constants that had not been previously measured were allowed to float. We assumed pointed end elongation was negligible. To limit the number of floated parameters in a given simulation, we first conducted polymerization assays with actin alone at a range of concentrations (2–6 μM), and then determined a reaction pathway and rate constants that could accurately describe spontaneous nucleation and polymerization of actin alone (see Supplementary Figure 2). The on rates for actin dimerization, trimerization, and tetramerization were fixed at $1.16 \times 10^7$ M$^{-1}$s$^{-1}$, the observed on rate for actin monomers binding to filament barbed ends. To simplify the models, steps that created a nucleus were considered irreversible and nuclei were modeled as catalysts that convert monomeric actin to filamentous actin, as previously described (*Helgeson and Nolen, 2013*; *Beltzner and Pollard, 2008*). We note that the best fits for spontaneous nucleation and elongation of actin filaments were obtained using a model in which either a dimer, trimer, or tetramer could serve as the nucleus. This pathway is distinct from models we previously used to simulate spontaneous nucleation and elongation in actin alone reactions (*Helgeson and Nolen, 2013*). To model reactions containing Dip1 and Arp2/3 complex (*Figure 3*), we fixed the rate constants for the spontaneous nucleation of actin at the values determined in reactions containing actin alone, except for $k_{-1}$, which was re-evaluated based on an 'actin alone' polymerization time course measured at the same time (in the same set) as the Dip1 and Arp2/3-containing reactions. We used the objective value, or the normalized mean square weighted sum of squares, as a measure of how well the model fits the experimental data. The mean square weighted sum of squares (MSS) was determined by the following equation: $MSS = \Sigma \omega * (x - y)2$ , where $\omega = \frac{1}{x^2}$ and $x$ and $y$ correspond to the experimental point in the dataset and simulated value, respectively.

## Yeast strain construction

*S. pombe* strains were constructed by PCR-based gene tagging and genetic crossing (see *Supplementary file 4*). Deletion cassettes with selectable markers were introduced into endogenous chromosomal loci by homologous recombination of PCR-amplified gene tagging modules (*Bähler et al., 1998*; *Wach et al., 1994*). Amplified modules were introduced into cells by lithium acetate transformation (*Keeney and Boeke, 1994*). Successful integrants were isolated based on selectable markers and confirmed by PCR and sequencing.

Fim1-mEGFP expressing cells were constructed in a previous study (*Wu and Pollard, 2005*). The *dip1Δ* strain was constructed by replacing the gene open reading frame with a *ura4*+ cassette amplified from KS-ura4 (*Bähler et al., 1998*). The C-terminal truncation of Wsp1 was constructed by replacing the sequence encoding the CA domain (aa 541–574) in the endogenous *wsp1* locus with the *stop codon-Tadh1-natMX6* cassette amplified from a custom-made pFA6a-GFP-natMX6 plasmid.

Strains combining two or more deletion or tagged alleles were constructed by standard genetic crosses on Malt Extract (ME) plates and tetrad dissection on YES plates followed by screening for wanted gene combinations by replica plating onto appropriate selective plates, microscopy, and PCR diagnostics.

## Live cell imaging

Imaging was performed on an UltraView VoX Spinning Disc Confocal System (PerkinElmer) mounted on a Nikon Eclipse Ti-E microscope with a 100×/1.4 NA PlanApo objective, equipped with a Hamamatsu C9100-50 EMCCD camera, and controlled by Volocity (PerkinElmer) software. Stably integrated *S. pombe* strains were grown to $OD_{595}$ = 0.2–1.0 in liquid YES medium (Sunrise) while shaking in the dark at 25°C over 2 days. For microscopy, cells from 0.5 to 1 mL of culture were collected by a brief centrifugation in a microfuge at 2000 g and 5 µL of partly resuspended pellet was placed onto a pad of 25% gelatin in EMM on a glass slide, covered by a coverslip and sealed with VALAP (1:1:1 vaseline:lanolin:paraffin mix). Samples were imaged after a 5 min incubation to allow for partial depolarization of actin patches. For time-lapse imaging, single-color images in a cell medial plane were taken every second for 60 s. Z-series of images spanning the entire cell width were captured at 0.4 µm intervals.

## Live cell image analysis

Image analysis was performed in ImageJ (National Institutes of Health). To make sure that different mutant backgrounds did not alter the expression levels of tagged proteins and that imaging conditions remained stable throughout an imaging session, for each dataset, we measured average background-subtracted whole-cell intensities, which correspond to the tagged protein expression levels. For each time series, we measured whole cell intensities of five cells, subtracted these values for either extracellular background or the intensities of untagged wild-type cells, and averaged the background-subtracted values at each time point. All Fim1-GFP strains within each dataset had similar whole-cell intensities.

The intensities and positions of fluorescent protein-tagged markers in individual endocytic actin patches were manually tracked throughout lifetime of each patch using a circular region of interest (ROI) with a 10-pixel diameter. Mean fluorescence intensities of patches were subtracted for mean cytoplasmic intensities measured in cell areas away from patches, and distances traveled by patches from the original positions were calculated. Time courses of background subtracted intensities and distances from origin for individual patches were aligned to the time of peak intensity (time = 0) and averaged at each time point.

Percent of internalized patches was measured by following fates of all patches present in frame 25 of time-lapse movies in a single medial plane. A patch was counted as internalized if it shifted from its original position by more than two pixels and, for each cell, the number of internalized patches was divided by the total number of analyzed patches. Percent internalization was measured and averaged from 10 wild-type, 10 *wsp1Δ* cells, 22 *dip1Δ*, and 20 *wsp1ΔCA dip1Δ* cells.

To measure patch density, all patches in 10 cells were counted in a z-series taken at a single time point. The area of the cell was measured at the medial focal plane and the patch density was calculated by dividing the number of patches by the cell area. Patch initiation rates (patches/µm²/s) were measured by counting patches that newly appeared during the first 20 s of time-lapse movies taken in a single medial cell plane and dividing the number of new patches by cell area and 20 s. The patches that were already present in frame 1 were excluded from the count.

## Statistical analysis

All fluorescence intensity, distance, patch initiation, and patch density values are displayed as mean ± SEM, with the exception of *Figure 1B*, which shows the mean ± standard deviation. The data are from single imaging experiment where all strains were imaged under identical conditions. The number of cells and patches analyzed are indicated in figure legends. The statistical significance for datasets of more than two conditions was tested by one-way ANOVA with Tukey's post-hoc test in GraphPad Prism. Both tests were two tailed and the significance values are reported in figures and figure legends.

## Acknowledgements

Research reported in this publication was supported by the National Institute of General Medical Sciences of the NIH under award number R35GM136319 (BJN) and T32 GM007759 (to CJB and LAH) and by the American Heart Association, grant no. 18PRE33960110 (CJB). We thank Andrew Wagner

for generating the *dip1Δ S. pombe* strain and for help with the two-color TIRF microscopy experiments.

## Additional information

### Funding

| Funder | Grant reference number | Author |
|---|---|---|
| National Institute of General Medical Sciences | R35GM136319 | Brad J Nolen |
| National Institute of General Medical Sciences | GM007759 | Connor John Balzer Luke A Helgeson |
| American Heart Association | 18PRE33960110 | Connor John Balzer |

The funders had no role in study design, data collection and interpretation, or the decision to submit the work for publication.

### Author contributions

Connor J Balzer, Conceptualization, Formal analysis, Investigation, Visualization, Methodology, Writing - original draft, Writing - review and editing; Michael L James, Formal analysis, Investigation, Methodology; Heidy Y Narvaez-Ortiz, Data curation, Formal analysis, Investigation, Writing - review and editing; Luke A Helgeson, Conceptualization, Investigation, Writing - review and editing; Vladimir Sirotkin, Resources, Formal analysis, Supervision, Methodology, Writing - review and editing; Brad J Nolen, Conceptualization, Formal analysis, Supervision, Funding acquisition, Investigation, Visualization, Writing - original draft, Project administration, Writing - review and editing

### Author ORCIDs

Luke A Helgeson ⓘ https://orcid.org/0000-0001-5112-2751
Vladimir Sirotkin ⓘ https://orcid.org/0000-0002-4938-9710
Brad J Nolen ⓘ https://orcid.org/0000-0002-0224-9980

### Decision letter and Author response

Decision letter https://doi.org/10.7554/eLife.60419.sa1
Author response https://doi.org/10.7554/eLife.60419.sa2

## Additional files

### Supplementary files

• Supplementary file 1. Summary of the fits of maximum polymerization rate data in *Figure 2* panel B,H and *Figure 3* panel F. The best-fit values for the maximum polymerization rate at saturation (Max. Poly. Rate$_{sat}$) and the concentration of Dip1 ($\mu$M) needed to reach half-saturated max polymerization rate for Dip1 alone or in the presence of Wsp1-VCA, GST-Wsp1-VCA, or Wsp1-CA. Data were fit to the following equation: Max poly rate = (max poly rate$_{max}\times$ [Dip1])/(K$_{1/2}$ + [Dip1]) + y-intercept. The y-intercept was set as the maximum polymerization rate in the absence of Dip1 for each condition.

• Supplementary file 2. Summary of values used in simulations of actin polymerization time courses in *Figure 3*. Rate constants were set based on previous published values or optimization of parameters using simple models generated in this study, as described in the main text and methods or indicated by the given reference. Units for k$_{on}$ are M$^{-1}$s$^{-1}$ except for reactions 4, 5, 6, and 11 where the units are s$^{-1}$. * indicates that a range of values for these parameters can yield a good fit to the experimental data (see *Figure 3—figure supplement 2*).

• Supplementary file 3. Comparison of optimized (fitted) values using single monomer-binding model to simulate polymerization time courses in the presence or absence of Wsp1. The table depicts a subset of *Supplementary file 2* reaction rate constants that were floated in the one

monomer-binding model to determine the best fit parameter values for datasets with Arp2/3 complex and Dip1 with or without Wsp1. Units for $k_{on}$ are $M^{-1}s^{-1}$ except for reaction 11, where units are $s^{-1}$. * indicates that a range of values can yield a good fit to the experimental data (see *Figure 3— figure supplement 2*).

- Supplementary file 4. *Schizosaccharomyces pombe* and *Saccharomyces cerevisiae* strains used in this study.
- Transparent reporting form

### Data availability

All data generated and analyzed in this study have been included in the manuscript and supporting source data files. An individual source data file has been provided for Figures 1, 2, 3 and 4.

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
