## [Decision Letter]

**Acceptance summary:**

This work illustrates very well the complexity of the mechanisms used by cells to generate new networks of actin filaments in a controlled manner. It also shows again clearly that actin nucleators and nucleation promoting factors should not only be studied individually, but also synergistically.

**Decision letter after peer review:**

Thank you for submitting your article "Synergy between Wsp1 and Dip1 may initiate assembly of endocytic actin networks" for consideration by *eLife*. Your article has been reviewed by three peer reviewers, including Alphée Michelot as the Reviewing Editor and Reviewer #1, and the evaluation has been overseen by Suzanne Pfeffer as the Senior Editor.

The reviewers have discussed the reviews with one another and the Reviewing Editor has drafted this decision to help you prepare a revised submission.

Summary:

This work investigates how two proteins, Dip1 and Wsp1, collaborate to activate the Arp2/3 complex and nucleate actin filaments. This lab had previously published three articles describing the activation mechanism of the Arp2/3 complex by Dip1. Because Dip1 was shown as sufficient to generate linear actin filaments, these studies led the authors to consider a model where the function of Dip1 would be the initiation of actin filament primers, from which Wsp1 and Arp2/3 could branch daughter filaments in order to form a complete branched networks.

This work reveals now that this model was partially incomplete, as Dip1 and Wsp1 synergize to activate Arp2/3 complex for the initiation of linear seed filaments. They built an extensive simulation of Dip1-induced Arp2/3 nucleation, which shows that the ability of Wsp1 to provide the first actin monomer(s) of the newly nucleated filament is part of the Dip1-Wsp1 interaction. Their model further suggests that Wsp1 potentially influences Dip1 binding and activation of Arp2/3.

Essential revisions for this paper:

1) One of the main concerns is whether the number of new endocytic sites and initiation rates are good indicators of the efficiency of actin nucleation in yeast. Now that we know that the cellular context has limited resources, where actin assembly is limited by the availability of G-actin, we cannot ignore the possibility that these two parameters may be primarily indicators of G-actin availability. As mutants cells with a lower number of actin patches also have brighter Fim1-GFP signals and longer lifetimes, it is likely that actin networks are more stable and G-actin is depleted, which could explain the decreased initiation of new endocytic sites. The authors should therefore provide additional information to rule out this possibility. This would also include phalloidin labeling to visualize actin cables in these strains in order to obtain a more picture of actin organization in mutant cells.

Increased data set in Figure 1 is also necessary. For example, in Figure 1F, the double mutant seems to present two populations of cells with distinct dynamics of patch internalization, but we do not know if this is true or if this is the effect of analyzing a very small number of cells (sometimes only 6 cells).

2) The simulation of the authors strongly suggests that Wsp1 also helps Dip1 to bind and activate Arp2/3. To complete their model, the authors should also test whether Wsp1 and Dip1 synergize in stimulating the short pitch conformation of Arp2/3, for example by using the crosslinking assay they applied in Wagner et al., 2013.

3) Experiment of Figure 4B is confusing, because the legend does not mention the presence and concentration of Wsp1 in these reactions. It is therefore difficult to analyze the shape of these curves compared to previous experiments. It would be also useful to explain in more details the logic that was chosen behind kinetic rate adjustments in this experiment.

i) We agree that varying dissociation constants is equivalent to modulating affinity constants, but from a purely kinetic point of view, adjusting association constants would also be meaningful and could potentially bring different results. Why was this possibility not considered?

ii) The Results section indicates that unfitted rate constants were fixed at values determined for reactions without Wsp1. However, Supplementary file 3 and Figure 3—figure supplement 2 indicate that few parameters in the Dip1 Alone dataset are not sensitive to variations, suggesting that affinity constants of these reactions are not important parameters. Could the authors detail more precisely which parameters can be precisely determined without Wsp1, and which ones are not sensitive and can be let free for a first fit with Wsp1? Once this first fit is performed, which parameters must be still changed to obtain a better fit in the presence of Wsp1?

iii) Is it possible to find additional experimental conditions to determine more precisely which of the kinetic parameters in reactions 9, 10 and 11 are changed or not changed in the presence of Wsp1?

4) TIRFM and fluorescence spectroscopy experiments are carried out with actin monomers that can either spontaneously nucleate filaments or elongate Dip1-Arp2/3 complex nucleated seeds. Moreover, addition of Wsp1-VCA in a relatively high molar ratio to actin (1:1.5) in the TIRFM assays may deplete actin monomers and favor Dip1-Arp2/3 mediated nucleation. The authors should confirm that these effects are negligible by using profilin-actin complexes in combination with a C-terminal fragment of Wsp1 comprising the proline-rich and VCA domains in a TIRFM and a pyrene assay.

5) Figure 2F: The Y-axis is labeled "% pointed ends bound by Dip1". It is unclear whether the authors refer to the % of filaments that are nucleated by Dip1 or whether they refer to the number of pointed ends that is decorated by 568-Dip1. The latter implies that this graph shows filaments nucleated by Dip1 as well as filaments that were bound by Dip1 after nucleation. If this is the case, the authors should show both the % of filaments nucleated by Dip1 as well as the % of filaments bound by Dip1.

Revisions expected in follow-up work:

The reviewers agree that this study would benefit from directly demonstrating the formation of a complex comprising Dip1 and Wsp1 during activation of the Arp2/3 complex. Ideally, a TIRFM experiment with labelled Wsp1 and Dip1 would show this interaction unambiguously. If this experiment is too challenging, the authors should consider using fluorescence anisotropy, or perhaps ITC with unlabeled proteins.

---

## [Author Response]

Essential revisions for this paper:1) One of the main concerns is whether the number of new endocytic sites and initiation rates are good indicators of the efficiency of actin nucleation in yeast. Now that we know that the cellular context has limited resources, where actin assembly is limited by the availability of G-actin, we cannot ignore the possibility that these two parameters may be primarily indicators of G-actin availability. As mutants cells with a lower number of actin patches also have brighter Fim1-GFP signals and longer lifetimes, it is likely that actin networks are more stable and G-actin is depleted, which could explain the decreased initiation of new endocytic sites. The authors should therefore provide additional information to rule out this possibility. This would also include phalloidin labeling to visualize actin cables in these strains in order to obtain a more picture of actin organization in mutant cells.

We thank the reviewers for bringing this to our attention. We believe previously published observations can address these concerns. First, our collaborative study with the Kovar lab (Suarez et al. Dev Cell 2015) showed that the actin patch initiation rate is not primarily an indicator of G-actin availability. Specifically, this study showed that overexpression of actin did not change the rate of actin patch initiation. Furthermore, data from Basu and Chang (Current Biology, 2013) strongly suggest that instead of decreasing the cytoplasmic concentration of actin monomers, Dip1 deletion actually increases it. Figure 1A in that paper shows that more polymerized actin is present in the actin cables in the Dip1 deletion strain than in the wild type strain (as visualized by staining with Alexa-488 phalloidin). The simplest interpretation of this result is that while individual actin patches contain more actin in the Dip1 deletion strain, there are fewer patches, so the cytoplasmic concentration of actin monomers increases, consequently increasing the amount of actin monomers available to assemble into cables.

To provide additional evidence that a hypothetical decrease in the cytoplasmic concentration of actin monomers suggested by reviewers cannot explain the decreased rate of actin patch initiation rate in the mutant strains, we expressed GFP-actin from the *arp3* promoter in an exogenous locus to determine the ratio of GFP-actin in the patches to total GFP-actin in the cell in the mutant versus wild type strains (note that previous experiments showed that GFP-actin is incorporated into actin patches but not actin cables (Wu and Pollard, 2005, Sirotkin, et al., 2010)). This new data showed that the ratio of actin monomer concentration in the patches versus cytoplasm is decreased in each of the mutant strains (*dip1Δ*, *wsp1Δ*CA *, dip1Δ wsp1Δ*CA) compared to the wild-type strain (see Figure 1—figure supplement 1). This is the opposite of what would be expected if the decreased actin patch initiation rates were due to excess incorporation of actin into the patches decreasing cytoplasmic actin monomer concentrations.

Increased data set in Figure 1 is also necessary. For example, in Figure 1F, the double mutant seems to present two populations of cells with distinct dynamics of patch internalization, but we do not know if this is true or if this is the effect of analyzing a very small number of cells (sometimes only 6 cells).

We increased the number of cells analyzed in panels D, E, and F in Figure 1 as requested. We note that we are careful not to make any claims about whether the double mutant shows one or multiple distinct populations. We only claim that the percent of internalizing patches is decreased compared to the wild type in the strains containing the *wsp1ΔCA* mutation, which we hope the reviewers agree is adequately supported by the data.

2) The simulation of the authors strongly suggests that Wsp1 also helps Dip1 to bind and activate Arp2/3. To complete their model, the authors should also test whether Wsp1 and Dip1 synergize in stimulating the short pitch conformation of Arp2/3, for example by using the crosslinking assay they applied in Wagner et al., 2013.

We carried out these experiments and included the new data as Figure 4—figure supplement 2. The data show that Wsp1 and Dip1 cooperate to stimulate the short pitch conformation, both in the presence or absence of actin monomers. We added a description of these new results in the revised Results section:

“We wondered whether the activating step(s) represented by k_11_ might be one of the steps accelerated in the presence of both Dip1 and Wsp1. […] Dip1 also increased short pitch crosslinking when added to a reaction containing Wsp1 and actin monomers.”

3) Experiment of Figure 4B is confusing, because the legend does not mention the presence and concentration of Wsp1 in these reactions. It is therefore difficult to analyze the shape of these curves compared to previous experiments.

We thank the reviewers for catching this oversight. We note that in the original version submitted for editorial consideration the presence of Wsp1 was not mentioned in the figure legend for Figure 4. However, in the full version submitted for review we had corrected this error. The corrected legend is also present in the revised version we are submitting in response to the reviews.

It would be also useful to explain in more details the logic that was chosen behind kinetic rate adjustments in this experiment.i) We agree that varying dissociation constants is equivalent to modulating affinity constants, but from a purely kinetic point of view, adjusting association constants would also be meaningful and could potentially bring different results. Why was this possibility not considered?

We repeated the analysis in Figure 4 to include simulations in which the association rate constants were varied. The new analysis is included as Figure 4—figure supplement 1. The conclusions from these simulations are identical to those in which the k_off_ was varied. First, increasing the affinity of actin monomers for the Dip1-Arp2/3 assembly (through increased k_on_) cannot account for the acceleration of polymerization we see in the assays. Second, varying the rate constants for any two of the three reactions (reaction 9, 10 or 11) is sufficient to give a good fit to the polymerization time courses. We added a sentence to the Results section to describe these results.

ii) The Results section indicates that unfitted rate constants were fixed at values determined for reactions without Wsp1. However, Supplementary file 3 and Figure 3—figure supplement 2 indicate that few parameters in the Dip1 Alone dataset are not sensitive to variations, suggesting that affinity constants of these reactions are not important parameters. Could the authors detail more precisely which parameters can be precisely determined without Wsp1, and which ones are not sensitive and can be let free for a first fit with Wsp1? Once this first fit is performed, which parameters must be still changed to obtain a better fit in the presence of Wsp1?

We thank the reviewers for prompting us to clarify these results. Our previous explanation of the meaning of the asterisks marking rate constants k_-10_ and k_11_ in Supplementary file 2 was inadequate. These rate constants (k_-10_ and k_11_) are highly correlated. Therefore, while there are many possible values for these rate constants that are consistent with the data, fixing one of these rate constants sets the optimal value for the other. This means that even though we cannot determine their precise values, both of these rate constants are important parameters in the simulation. To clarify this, we changed the plots in Figure 3—figure supplement 2D and E. These panels previously showed only the best objective value for simulations in which either k_11_ or k_-10_ was varied over a range of values and the other rate constants were floated. The new versions of these panels also show the corresponding value of k_-10_ (panel D) or k_11_ (panel E) that gave the best objective value. The parameter scans in panels D and E show that as k_-10_ increases, k_11_ must increase for the simulations to fit the data. Conversely, as k_11_ increases, k_-10_ must increase for the simulations to fit the data. These plots (C-E) also show that while many combinations of k_-10_ and k_11_ can produce simulations that fit the data well, the value of k_-9_ is well determined – values significantly less or greater than 9.8 s^-1^ show poorer fits (greater objective values). We note that though this parameter is well-determined in the simulation, whether it approximates the off rate of Dip1 for isolated Arp2/3 complex depends on the validity of several assumptions we made in making the model. This includes, for instance, that the on rate of Dip1 binding to Arp2/3 complex is 1x10^6^ M^-1^s^-1^ and that Dip1 does not dissociate from the Dip1-Arp2/3-actin assembly.

iii) Is it possible to find additional experimental conditions to determine more precisely which of the kinetic parameters in reactions 9, 10 and 11 are changed or not changed in the presence of Wsp1?

We think there may be ways to better determine the changes in the reaction pathway and the rate constants for individual elementary reaction steps that allow for acceleration of the reaction in the presence of WASP and Dip1, but we don’t believe that we will be able to answer these questions by changing the experimental conditions in the pyrene actin polymerization assay. It is important to note that the kinetic scheme we used to understand synergy is a simplified (implicit) one. To go far beyond our current understanding of the kinetic mechanism of synergistic activation, it is likely we will have to build an explicit model that includes many more reaction steps, including binding of Wsp1 to each of its two sites on Arp2/3 complex, dissociation of Dip1 from Arp2/3 bound to actin monomers, multiple conformational changes of Arp2/3 complex, etc. Such a model will be an important starting point but will require significant time and effort to develop and test.

However, we note that the revised draft of the manuscript includes in the Results section new evidence that provides insight into the possible rate enhancing steps. Specifically, the results of the new short pitch crosslinking assays suggest that Wsp1 and Dip1 together more efficiently stimulate the short pitch conformation than either NPF alone. Therefore, the kinetic step(s) represented by k_11_ may be accelerated in synergistic activation.

4) TIRFM and fluorescence spectroscopy experiments are carried out with actin monomers that can either spontaneously nucleate filaments or elongate Dip1-Arp2/3 complex nucleated seeds. Moreover, addition of Wsp1-VCA in a relatively high molar ratio to actin (1:1.5) in the TIRFM assays may deplete actin monomers and favor Dip1-Arp2/3 mediated nucleation. The authors should confirm that these effects are negligible by using profilin-actin complexes in combination with a C-terminal fragment of Wsp1 comprising the proline-rich and VCA domains in a TIRFM and a pyrene assay.

The reviewers bring up the important point that the increased fraction of Dip1-bound actin filaments in TIRF reactions with Dip1 could be misinterpreted as synergy between Dip1 and Wsp1 when in reality this value increased because quantity of actin filaments spontaneously nucleated decreases at high Wsp1 concentrations. This concern prompted us to re-analyze our data. We found that multiple aspects of the data in Figure 2 exclude the possibility that the increased number of Dip1-bound ends is due to reduced spontaneous nucleation.

a) We measured the total number of spontaneously nucleated filaments at t = 150 s in reactions with and without Wsp1 (VCA or GST-VCA). We found that while both GST-tagged and monomeric VCA each decreased the number of spontaneously nucleated filaments by ~1/2 (new Figure 2—figure supplement 1), GST-VCA did not significantly increase the percent of filaments with Dip1 bound under these conditions (at 100nM Arp2/3 complex – see Figure 2D). In contrast VCA potently increased the number of Dip1-bound filament ends at these concentrations (Figure 2D). Because monomeric and dimerized VCA inhibit spontaneous nucleation to the same extent, the potent increase in Dip1-bound filaments with VCA cannot be explained by a decrease in the rate of spontaneous nucleation.

b) Based on our measurements of the number of spontaneously nucleated filaments with and without Wsp1, we normalized the % pointed ends bound to account for the reduction in spontaneous nucleation due to the presence of Wsp1 in Figure 2D and F. The data show that even when accounting for the reduced spontaneous nucleation, both Wsp1-VCA (Figure 2D) and GST-Wsp1-VCA (Figure 2F) significantly increase the percentage of linear filament ends with Dip1 bound.

c) We measured the number of Dip1 bound ends at time = 150 s and expressed it as a raw number rather than a percent of total linear filament ends. The raw number of Dip1-bound ends in a field of view is independent of the rate of spontaneous nucleation, so this number provides a direct readout of the influence of Wsp1 on Dip1-Arp2/3-mediated nucleation. These analyses showed that both Wsp1-VCA (Figure 2—figure supplement 1A) and GST-Wsp1-VCA (Figure 2—figure supplement 1C) significantly increase the total number of Dip1 bound ends in a field. We note that there is greater variation in the number of Dip1 bound ends than in the percent of Dip1 bound ends. This variation arises from small differences in the progress of the reaction at equivalent measured timepoints, since there is a correlation between the total number of linear filaments and the total number of Dip1 bound filament ends.

We note that we also included a new video (Video 5) that shows a comparison of actin polymerization assays observed by TIRF microscopy for reactions containing Oregon-green actin, Arp2/3 complex, and Alexa568-Dip1 with or without Wsp1-VCA.

5) Figure 2F: The Y-axis is labeled "% pointed ends bound by Dip1". It is unclear whether the authors refer to the % of filaments that are nucleated by Dip1 or whether they refer to the number of pointed ends that is decorated by 568-Dip1. The latter implies that this graph shows filaments nucleated by Dip1 as well as filaments that were bound by Dip1 after nucleation. If this is the case, the authors should show both the % of filaments nucleated by Dip1 as well as the % of filaments bound by Dip1.

Dip1 is not attached to the surface of the cover slip. Therefore, the only nucleation events that are directly observed are those in which Dip1 fortuitously adsorbs to the surface and a filament can be seen elongating from it, which is a small % of the events (~1%, see Balzer, et al., 2018). Most Dip1 bound filaments are nucleated in solution and then fall to the surface. However, we showed previously that under these conditions the vast majority of Dip1-bound filaments (~97%) are filaments that were nucleated by Dip1-bound Arp2/3 complex rather than filaments that spontaneously nucleated and then bound Arp2/3 and Dip1 (Balzer, et al., 2018). This is because Dip1 binds very weakly to Arp2/3 complex on the end of spontaneously nucleated actin filaments. Therefore the number of linear filaments decorated with Dip1 is nearly identical to the number of Dip1-Arp2/3-nucleated actin filaments. To clarify this, we edited the figure legend for Figure 2:

“D. Quantification of the percentage of linear actin filament pointed ends bound by 568-Dip1 two minutes and thirty seconds into actin polymerization assays in C. […] The total number of linear actin filaments was corrected to account for a ~2-fold decrease in the number of spontaneously nucleated actin filaments caused by inhibition of spontaneous nucleation by GST-Wsp1-VCA or Wsp1-VCA (See Figure 2—figure supplement 1).”

Note that we also clarified that the percentage of pointed ends calculation includes only linear actin filaments and not branches.

Revisions expected in follow-up work:The reviewers agree that this study would benefit from directly demonstrating the formation of a complex comprising Dip1 and Wsp1 during activation of the Arp2/3 complex. Ideally, a TIRFM experiment with labelled Wsp1 and Dip1 would show this interaction unambiguously. If this experiment is too challenging, the authors should consider using fluorescence anisotropy, or perhaps ITC with unlabeled proteins.

We agree that demonstrating formation of a complex between Arp2/3, Wsp1 and Dip1 would be a good addition to the paper. To this end, we used GST-Wsp1 to pull down Dip1 in solutions with or without Arp2/3 complex. We could not detect a significant amount of binding in these reactions using wild type SpArp2/3 complex. We suspect that this is because Dip1 binds relatively weakly to Arp2/3 complex and the interaction is difficult to detect by pull down even if Dip1 is directly attached to the bead (see Wagner et al., 2013). However, when we used the Arp3ΔC complex (in which the C-terminus of Arp3 is deleted) for the assays, Dip1 could be pulled down if Arp2/3 complex was present (but not with Wsp1 alone), indicating formation of a Dip1-Arp2/3-Wsp1 assembly. The Arp3ΔC complex binds ~7 fold more tightly to Wsp1 than wild type Arp2/3 complex, so we expect that this mutant makes detection of the Wsp1-Arp2/3-Dip1 assembly possible by increasing saturation of the Wsp1 beads with the complex, thereby increasing the concentration on the beads of the Wsp1-Arp2/3 assembly enough to detect the relatively weak Arp2/3-Dip1 interaction.

We added the pulldown data as part of the new Figure 4—figure supplement 2 and added the following text to the Results section to incorporate this new data into the manuscript:

“Given that co-stimulation of the short pitch conformation by Wsp1 and Dip1 would require simultaneous binding of Arp2/3 complex by the two NPFs, we also asked if Wsp1 and Dip1 co-bind to Arp2/3 complex. […] Detection of the Dip1-Arp2/3-Wsp1 assembly with this method required using a mutant Arp2/3 complex that binds with increased affinity to Wsp1, presumably because the Dip1-Arp2/3 interaction is weak (Wagner et al., 2013).”